# Interferon receptor-deficient mice are susceptible to eschar-associated rickettsiosis

**Thomas P Burke[1]\***, **Patrik Engström[1]**, **Cuong J Tran[1,2]**, **Ingeborg M Langohr[3]**, **Dustin R Glasner[2†]**, **Diego A Espinosa[2‡]**, **Eva Harris[2]**, **Matthew D Welch[1]\***

[1]Molecular and Cell Biology, University of California, Berkeley, Berkeley, United States; [2]Division of Infectious Disease and Vaccinology, School of Public Health, University of California, Berkeley, Berkeley, United States; [3]Department of Pathobiological Sciences, Louisiana State University, Baton Rouge, United States

**\*For correspondence:**
tburke@berkeley.edu (TPB);
welch@berkeley.edu (MDW)

**Present address:** [†]Department of Laboratory Medicine, University of California, San Francisco, San Francisco, United States; [‡]Metagenomi, Emeryville, United States

**Competing interest:** The authors declare that no competing interests exist.

**Abstract** Arthropod-borne rickettsial pathogens cause mild and severe human disease worldwide. The tick-borne pathogen *Rickettsia parkeri* elicits skin lesions (eschars) and disseminated disease in humans; however, inbred mice are generally resistant to infection. We report that intradermal infection of mice lacking both interferon receptors (*Ifnar1*$^{-/-}$;*Ifngr1*$^{-/-}$) with as few as 10 *R. parkeri* elicits eschar formation and disseminated, lethal disease. Similar to human infection, eschars exhibited necrosis and inflammation, with bacteria primarily found in leukocytes. Using this model, we find that the actin-based motility factor Sca2 is required for dissemination from the skin to internal organs, and the outer membrane protein OmpB contributes to eschar formation. Immunizing *Ifnar1*$^{-/-}$;*Ifngr1*$^{-/-}$ mice with *sca2* and *ompB* mutant *R. parkeri* protects against rechallenge, revealing live-attenuated vaccine candidates. Thus, *Ifnar1*$^{-/-}$;*Ifngr1*$^{-/-}$ mice are a tractable model to investigate rickettsiosis, virulence factors, and immunity. Our results further suggest that discrepancies between mouse and human susceptibility may be due to differences in interferon signaling.

## Introduction

Obligate cytosolic bacterial pathogens in the family Rickettsiaceae are a diverse group of arthropod-borne microbes that cause severe human disease worldwide, including spotted fever, scrub typhus, and typhus (**Bonell et al., 2017**; **Fang et al., 2017**; **Sahni et al., 2019**). Human disease caused by the tick-borne spotted fever group (SFG) pathogen *Rickettsia parkeri* is characterized by a necrotic, dry skin lesion (eschar) at the infection site, as well as generalized rash, headache, fatigue, and fever (**Paddock et al., 2008**). There is no approved vaccine for *R. parkeri* or for the more virulent rickettsial pathogens that can cause fatal or latent disease (**Osterloh, 2017**). Moreover, many critical aspects of disease caused by obligate cytosolic bacterial pathogens, including the mechanisms of virulence and immunity, remain unknown. *R. parkeri* can be handled under biosafety level 2 (BSL2) conditions, and characterizing mouse models to recapitulate major features of human disease would enhance research efforts into understanding rickettsial pathogens and disease (**Osterloh, 2017**; **Grasperge et al., 2012**; **Sunyakumthorn et al., 2013**).

*R. parkeri* is genetically similar to the more virulent human pathogens *R. rickettsii* and *R. conorii* (**Roux and Raoult, 2000**; **Goddard, 2009**), and it can be handled under BSL2 conditions. Moreover, mutants can be generated using transposon mutagenesis (**Reed et al., 2014**; **Lamason et al., 2018**), and small rodents including mice have been found as seropositive for *R. parkeri* in the wild (**Moraru et al., 2013a**; **Moraru et al., 2013b**; **Krawczak et al., 2016**; **Barbieri et al., 2019**). Thus, a mouse model for *R. parkeri* that recapitulates key features of human infection would greatly enhance

**eLife digest** Tick bites allow disease-causing microbes, including multiple species of *Rickettsia* bacteria, to pass from arthropods to humans. Being exposed to *Rickettsia parkeri*, for example, can cause a scab at the bite site, fever, headache and fatigue.

To date, no vaccine is available against any of the severe diseases caused by *Rickettsia* species. Modelling human infections in animals could help to understand and combat these illnesses. *R. parkeri* is a good candidate for such studies, as it can give insight into more severe *Rickettsia* infections while being comparatively safer to handle. However, laboratory mice are resistant to this species of bacteria, limiting their use as models.

To explore why this is the case, Burke et al. probed whether an immune mechanism known as interferon signalling protects laboratory rodents against *R. parkeri*. During infection, the immune system releases molecules called interferons that stick to 'receptors' at the surface of cells, triggering defense mechanisms that help to fight off an invader.

Burke et al. injected *R. parkeri* into the skin of mice that had or lacked certain interferon receptors, showing that animals without two specific receptors developed scabs and saw the disease spread through their body. Further investigation showed that two *R. parkeri* proteins, known as OmpB or Sca2, were essential for the bacteria to cause skin lesions and damage internal organs.

Burke et al. then used *R. parkeri* that lacked OmpB or Sca2 to test whether these modified, inoffensive microbes could act as 'vaccines'. And indeed, vulnerable laboratory mice which were first exposed to the mutant bacteria were then able to survive the 'normal' version of the microbe.

Together, this work reveals that interferon signalling protects laboratory mice against *R. parkeri* infections. It also creates an animal model that can be used to study disease and vaccination.

investigations into understanding rickettsial disease. However, inbred mice including C57BL/6 and BALB/c develop no or minor skin lesions upon intradermal (i.d.) infection with *R. parkeri* (*Grasperge et al., 2012*). Strategies to develop a mouse model, including infecting Toll-like receptor 4 (TLR4)-deficient mice (*Grasperge et al., 2012*), delivering high doses of bacteria (*Londoño et al., 2019*), and delivering *R. parkeri* via a tick vector (*Saito et al., 2019*), have been examined. However, a model that recapitulates eschar formation and dissemination via needle inoculation with low doses of *R. parkeri* has remained elusive. A mouse model for *R. parkeri* that mimics eschar formation and disseminated disease in C57BL/6 mice, the most widely used genetic background with a large variety of available mutants, would aid investigations into rickettsial virulence mechanisms, the host response to infection, and spotted fever disease.

Toward better understanding the host response to *R. parkeri* infection, we recently investigated the relationship between *R. parkeri* and interferons (IFNs), which are ubiquitous signaling molecules of the innate immune system that mobilize the cytosol to an antimicrobial state. Type I IFN (IFN-I) generally restricts viral replication, whereas IFN-γ generally restricts intracellular bacterial pathogens (*Raniga and Liang, 2018*; *Billiau and Matthys, 2009*; *Meunier and Broz, 2016*). We observed that mice lacking either gene encoding the receptors for IFN-I (*Ifnar1*) or IFN-γ (*Ifngr1*) are resistant to i.v. infection with *R. parkeri*, whereas double knockout (DKO) *Ifnar1*[-/-];*Ifngr1*[-/-] mice succumb (*Burke et al., 2020*). This demonstrates that IFNs redundantly protect against systemic *R. parkeri*. However, the i.v. infection route does not recapitulate eschar formation and i.d. infection may more closely mimic infection by tick bite. Further investigations into whether IFNs redundantly protect against *R. parkeri* in the skin may improve the mouse model for SFG *Rickettsia*.

A robust mouse model would facilitate investigations into conserved rickettsial virulence factors, whose role in pathogenesis in vivo remain poorly understood. One virulence mechanism shared by divergent cytosolic bacterial pathogens including *Rickettsia, Listeria, Burkholderia, Mycobacterium*, and *Shigella* species is the ability to undergo actin-based motility, which facilitates cell to cell spread (*Choe and Welch, 2016*; *Lamason and Welch, 2017*). However, the pathogenic role for many actin-based motility factors in vivo remains unknown. *R. parkeri* actin-based motility differs from that of other pathogens in that it occurs in two phases. The first phase requires the RickA protein, which elicits actin-based motility by activating the host Arp2/3 complex (*Gouin et al., 2004*; *Jeng et al., 2004*); however, RickA is dispensable for cell to cell spread in vitro (*Reed et al., 2014*). The second phase

requires the Sca2 protein (*Reed et al., 2014*; *Kleba et al., 2010*), which mimics eukaryotic formins to directly nucleate and elongate actin filaments (*Reed et al., 2014*; *Haglund et al., 2010*). Sca2 is required for efficient cell to cell spread, although it is not required for replication in epithelial cells or for avoiding antimicrobial autophagy (*Reed et al., 2014*; *Kleba et al., 2010*; *Haglund et al., 2010*; *Engström et al., 2019*). *sca2* mutant *R. rickettsii* elicit reduced fever in guinea pigs when compared with wild-type (WT) *R. rickettsii* (*Kleba et al., 2010*), yet the explanation for reduced fever and the pathogenic role for Sca2 in vivo remains unclear. Additionally, Sca2 is not essential for dissemination of *R. parkeri* within ticks (*Harris et al., 2018*).

A second virulence strategy employed by intracellular pathogens is the ability to avoid autophagy, which for *R. parkeri* requires the abundant, conserved outer membrane protein B (OmpB) (*Engström et al., 2019*; *Engström et al., 2021*). *ompB* mutant *R. parkeri* are ubiquitylated and restricted by antimicrobial autophagy in mouse macrophages, and OmpB is important for *R. parkeri* infection of internal organs in WT mice and for causing lethal disease in IFN receptor-deficient mice after i.v. infection (*Burke et al., 2020*; *Engström et al., 2019*). However, the role for *R. parkeri* OmpB upon i.d. infection remains unknown. An improved mouse model may improve our understanding of how conserved virulence factors including Sca2 and OmpB enhance rickettsial pathogenesis.

Here, we use IFN receptor-deficient mice to examine the effects of i.d. inoculation of *R. parkeri*, mimicking the natural route of infection. We observe skin lesions that appear similar to human eschars, as well as disseminated lethal disease with as few as 10 bacteria. Using this model, we find that Sca2 promotes dissemination and is required for causing lethality and that OmpB contributes to eschar formation and to lethal disease. We demonstrate that immunization with *sca2* or *ompB* mutant *R. parkeri* protects IFN receptor-deficient mice against subsequent challenge with WT bacteria, revealing live-attenuated vaccine candidates. Our study establishes a mouse model to investigate numerous aspects of *Rickettsia* pathogenesis, including eschar formation, virulence factors, and immunity. More broadly, this work also reveals that a potent, redundant IFN response protects mice from eschar-associated rickettsiosis.

## Results

### *Ifnar1*$^{-/-}$;*Ifngr1*$^{-/-}$ mice are susceptible to eschar-associated rickettsiosis

We sought to develop an i.d. murine infection model to better recapitulate the natural route of tick-borne *R. parkeri* infection. WT, single mutant *Tlr4*$^{-/-}$, *Ifnar1*$^{-/-}$, or *Ifngr1*$^{-/-}$, and DKO *Ifnar1*$^{-/-}$;*Ifngr1*$^{-/-}$ C57BL/6 J mice, as well as outbred WT CD-1 mice, were infected i.d. with $10^7$ WT *R. parkeri* and monitored over time. No or minor dermal lesions appeared at the site of infection in WT, single mutant *Tlr4*$^{-/-}$, *Ifnar1*$^{-/-}$, or *Ifngr1*$^{-/-}$ C57BL/6 J mice or CD-1 mice (*Figure 1*, *Figure 1—figure supplement 1a*). In contrast, DKO *Ifnar1*$^{-/-}$;*Ifngr1*$^{-/-}$ C57BL/6 J mice developed delimited skin lesions measuring >1 cm in diameter that were necrotic, hardened, non-pruritic, and surrounded by an indurated red halo (*Figure 1b*), similar to human eschars (*Figure 1c*; *Paddock et al., 2008*; *Paddock et al., 2004*; *Kaskas, 2014*; *Cragun et al., 2010*; *Herrick et al., 2016*). In some cases, tails of DKO *Ifnar1*$^{-/-}$;*Ifngr1*$^{-/-}$ or single mutant *Ifngr1*$^{-/-}$ mutant mice became inflamed after i.d. or i.v. infection (*Figure 1—figure supplement 1b*). These findings demonstrate that interferons redundantly control disease caused by *R. parkeri* in the skin and that i.d. infection of DKO *Ifnar1*$^{-/-}$;*Ifngr1*$^{-/-}$ mice recapitulates the hallmark manifestation of human disease caused by *R. parkeri*.

Our previous observations using the i.v. route revealed dose-dependent lethality in *Ifnar1*$^{-/-}$;*Ifngr1*$^{-/-}$ mice, with $10^7$ *R. parkeri* eliciting 100 % lethality and $10^5$ *R. parkeri* eliciting no lethality (*Burke et al., 2020*). *R. parkeri* are present in tick saliva at a concentration of approximately $10^4$ per 1 µl, and approximately $10^7$ *R. parkeri* are found in tick salivary glands (*Suwanbongkot et al., 2019*). However, the number of bacteria delivered from tick infestation likely varies depending on many factors, and we therefore sought to examine the effects of different doses of *R. parkeri* upon i.d. infection of *Ifnar1*$^{-/-}$;*Ifngr1*$^{-/-}$ mice. We observed skin lesion formation at all infectious doses, from $10^7$ to 10 bacteria (*Figure 1d*), suggesting that i.d. infection of *Ifnar1*$^{-/-}$;*Ifngr1*$^{-/-}$ mice elicits lesions with doses similar to what is delivered by tick infestation.

We next sought to quantitatively evaluate the effects of i.d. infection by monitoring animal weight, body temperature, the degree of lesion formation, and lethality. Intradermally infected DKO *Ifnar1*$^{-/-}$;*Ifngr1*$^{-/}$ mice lost significant body weight (*Figure 2a*, *Figure 2—figure supplement 1a*) and body

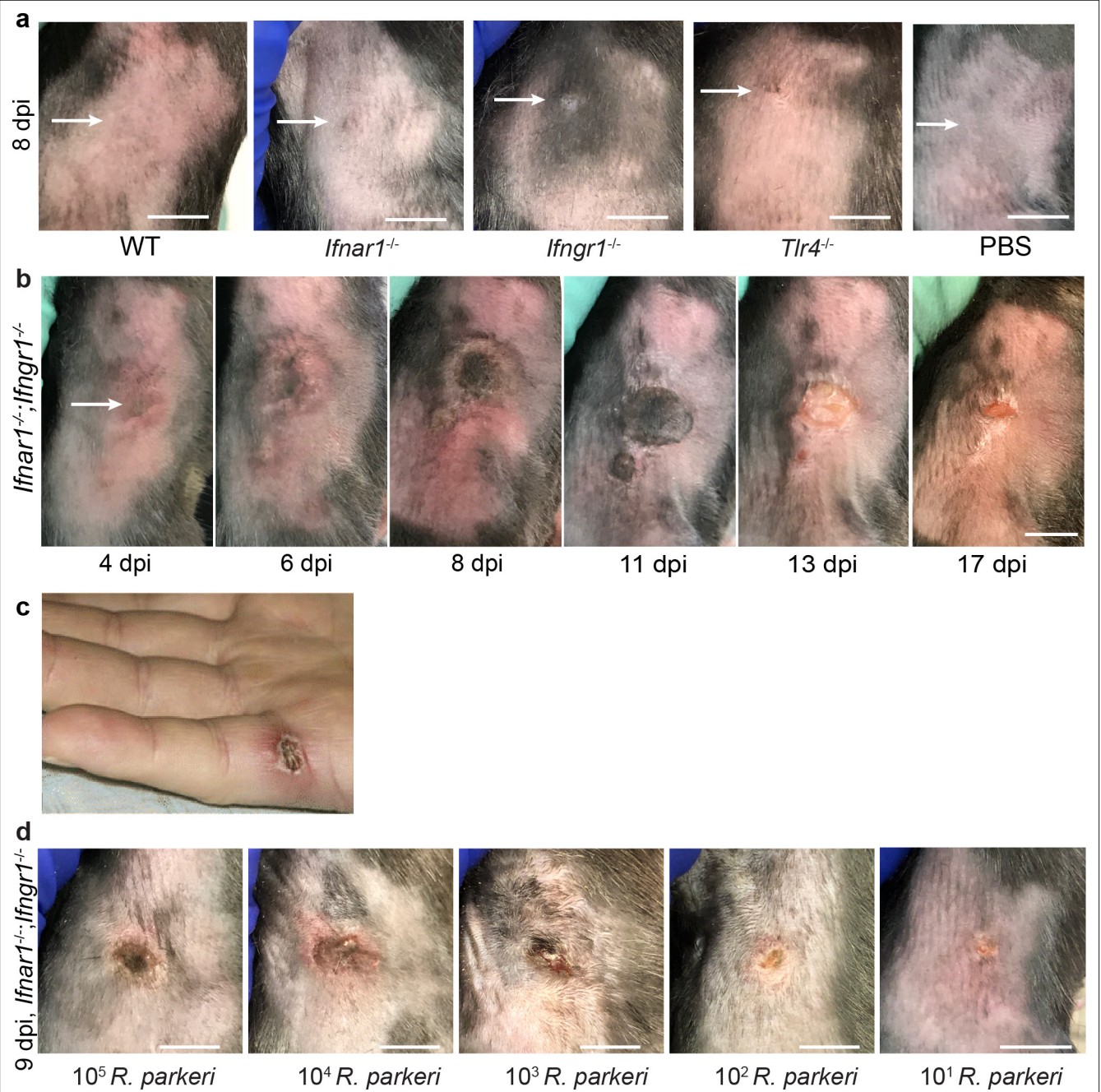

**Figure 1.** I.d. infection of *Ifnar1*[-/-];*Ifngr1*[-/-] mice with *R. parkeri* elicits skin lesions that are grossly similar to human eschars. (**a**) Representative images of WT, *Ifnar1*[-/-], *Ifngr1*[-/-], and *Tlr4*[-/-] single mutant mice, infected intradermally with $10^7$ WT *R. parkeri* PFU, at 8 d.p.i. WT mice were injected with PBS. White arrows indicate the infection site on the right flank of the mouse. Scale bar, 1 cm. Data are representative of three independent experiments. (**b**) Representative images of a DKO *Ifnar1*[-/-];*Ifngr1*[-/-] mouse after i.d. inoculation with $10^7$ *R. parkeri* PFU. Data are representative of three independent experiments. The white arrow indicates the injection site on the right flank of the mouse. Scale bar, 1 cm. (**c**) Gross pathology of a human *R. parkeri* infection, reproduced from **Figure 2a** of *Paddock et al., 2008*, with permission from Oxford Academic. It is not covered by the CC-BY 4.0 license and further reproduction of this panel would require permission from the copyright holder. (**d**) Representative images of *Ifnar1*[-/-];*Ifngr1*[-/-] mice infected intradermally with the indicated amounts of WT *R. parkeri* at 9 d.p.i. Scale bar, 1 cm. Data are representative of two independent experiments.

The online version of this article includes the following figure supplement(s) for figure 1:

**Figure supplement 1.** *Ifnar1*[-/-];*Ifngr1*[-/-] mice develop disseminated disease upon intradermal *R. parkeri* infection.

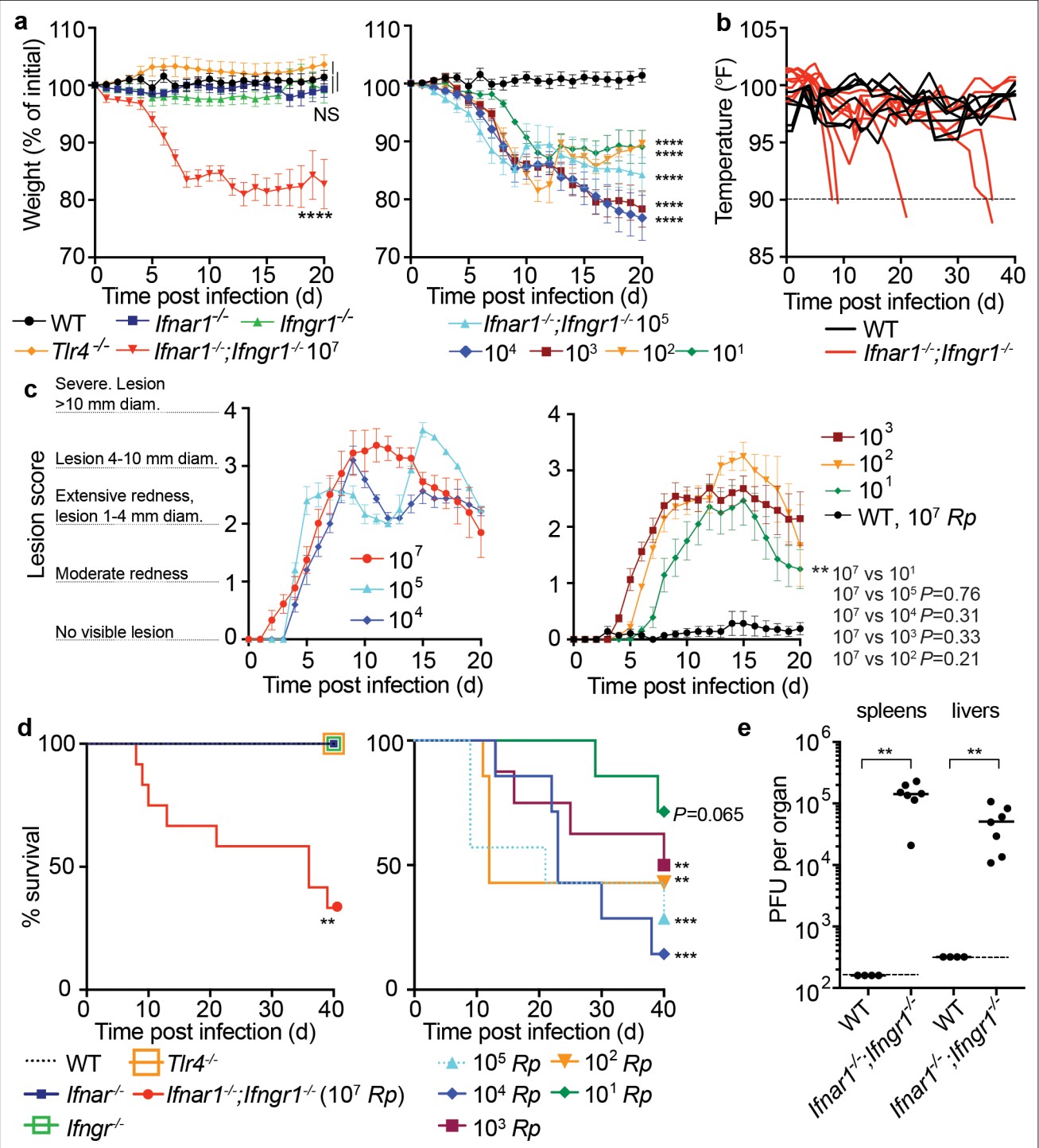

**Figure 2.** I.d. infection of *Ifnar1⁻/⁻;Ifngr1⁻/⁻* mice by *R. parkeri* elicits disseminated, lethal disease. (**a**) Weight changes over time in mice infected i.d. with *R. parkeri*. Data are shown as a percentage change to initial weight. In the left panel, all mice were infected with $10^7$ PFU of *R. parkeri*: n = 7 (WT), n = 11 (*Ifnar1⁻/⁻*), n = 7 (*Ifngr1⁻/⁻*), n = 9 (DKO *Ifnar1⁻/⁻;Ifngr1⁻/⁻*), and n = 4 (*Tlr4⁻/⁻*) individual mice. In the right panel, DKO *Ifnar1⁻/⁻;Ifngr1⁻/⁻* mice were infected with the indicated *R. parkeri* PFU: n = 7 ($10^5$), n = 7 ($10^4$), n = 8 ($10^3$), n = 7 ($10^2$), n = 7 ($10^1$) individual mice. WT data is the same in both panels. Data for each genotype are combined from two or three independent experiments. (**b**) Temperature changes over time in mice intradermally infected with $10^7$ *R. parkeri* PFU. Each line is an individual mouse. Mice were euthanized if their temperature fell below 90 °F, as indicated by the dotted line. Data are the combination of three independent experiments with n = 7 (WT) and n = 9 (*Ifnar1⁻/⁻;Ifngr1⁻/⁻*) individual mice. (**c**) Analysis of gross skin pathology after i.d. infection. *Ifnar1⁻/⁻;Ifngr1⁻/⁻* mice were infected with the indicated PFU of *R. parkeri* and monitored over time. WT mice were infected with $10^7$ *R. parkeri* PFU. Data are the combination of three independent experiments for WT and the $10^7$ dose in *Ifnar1⁻/⁻Ifngr1⁻/⁻* mice; data for all other doses are

*Figure 2 continued on next page*

*Figure 2 continued*

the combination of two independent experiments. n = 9 ($10^7$), n = 5 ($10^5$), n = 5 ($10^4$), n = 8 ($10^3$), n = 7 ($10^2$), n = 7 ($10^1$), and n = 7 (WT) individual mice. (**d**) Mouse survival after i.d. infection with *R. parkeri*. In the left panel, all mice were infected with $10^7$ *R. parkeri* PFU: n = 7 (WT), n = 11 (*Ifnar1*$^{-/-}$), n = 7 (*Ifngr1*$^{-/-}$), n = 4 *Tlr4*$^{-/-}$, and n = 12 (DKO *Ifnar1*$^{-/-}$;*Ifngr1*$^{-/-}$) individual mice. Data are the combination of three separate experiments for WT, *Ifnar1*, and DKO *Ifnar1*$^{-/-}$;*Ifngr1*$^{-/-}$ and two separate experiments for *Ifngr1*$^{-/-}$ and *Tlr4*$^{-/-}$. In the right panel, DKO *Ifnar1*$^{-/-}$;*Ifngr1*$^{-/-}$ mice were infected with the indicated amounts of *R. parkeri*. Data are the combination of two independent experiments: n = 7 ($10^5$), n = 7 ($10^4$), n = 8 ($10^3$), n = 7 ($10^2$), and n = 7 ($10^1$) individual mice. (**e**) Bacterial burdens in organs of intradermally infected WT and *Ifnar1*$^{-/-}$;*Ifngr1*$^{-/-}$ mice. Mice were intradermally inoculated with $10^7$ *R. parkeri*, and spleens and livers were harvested and plated for p.f.u. at 72 h.p.i. Dotted lines indicate the limit of detection. Data are the combination of two independent experiments. n = 4 (WT) and 7 (*Ifnar1*$^{-/-}$;*Ifngr1*$^{-/-}$) individual mice. Data in (**a**), (**c**) are the mean ± SEM. Statistical analyses in (**a**) used a two-way ANOVA where each group was compared to WT at t = 20 d.p.i. Statistical analyses in (**c**) used a two-way ANOVA at t = 20 d.p.i. Statistical analyses in (**d**) used a log-rank (Mantel−Cox) test to compare *Ifnar1*$^{-/-}$ to DKO *Ifnar1*$^{-/-}$;*Ifngr1*$^{-/-}$ at each dose. Statistical analysis in (**e**) used a two-tailed Mann−Whitney U test. NS, not significant; \*\*p<<0.01; \*\*\*p<0.001; \*\*\*\*p<0.0001.

The online version of this article includes the following source data and figure supplement(s) for figure 2:

**Source data 1.** I.d. infection of *Ifnar1*$^{-/-}$;*Ifngr1*$^{-/-}$ mice by *R. parkeri* elicits disseminated, lethal disease.

**Figure supplement 1.** *Ifnar1*$^{-/-}$ or *Ifngr1*$^{-/-}$ mice develop limited disease upon intradermal infection, and *Ifnar1*$^{-/-}$;*Ifngr1*$^{-/-}$ develop lesions of dose-dependent severity.

temperature (*Figure 2b*; animals were euthanized when body temperature fell below 90 °F / 32.2 °C) when compared with WT mice, whereas infected single mutant *Tlr4*$^{-/-}$, *Ifnar1*$^{-/-}$, or *Ifngr1*$^{-/-}$ mice did not. To evaluate lesion severity, we scored lesions upon infection with different doses of *R. parkeri*. Whereas $10^7$ bacteria elicited similar responses as $10^5$, $10^4$, $10^3$, and $10^2$ bacteria (*Figure 2c*), lesions were less severe when mice were infected with $10^1$ bacteria compared with $10^7$ bacteria. If mice survived, lesions healed over the course of approximately 15–40 days post infection (d.p.i.) at all doses (*Figure 2—figure supplement 1b*).

To investigate whether i.d. infection by *R. parkeri* caused lethal disease, we monitored mouse survival over time. Upon i.d. delivery of $10^7$ *R. parkeri*, 8 of 12 *Ifnar1*$^{-/-}$;*Ifngr1*$^{-/-}$ mice exhibited lethargy, paralysis, or body temperatures below 90°F, at which point they were euthanized, whereas delivery of the same dose of bacteria to WT and single mutant mice did not elicit lesions and all survived (*Figure 2d*). Lower doses of *R. parkeri* also elicited body weight loss (*Figure 2a*), body temperature loss (*Figure 2—figure supplement 1*), and lethal disease (*Figure 2d*) in *Ifnar1*$^{-/-}$;*Ifngr1*$^{-/-}$ mice. Degrees of lethality between different doses in *Ifnar1*$^{-/-}$;*Ifngr1*$^{-/-}$ mice were not significantly different from one another, and the cause of lethality in this model remains unclear. Nevertheless, these findings reveal that i.d. infection can cause lethal disease in *Ifnar1*$^{-/-}$;*Ifngr1*$^{-/-}$ mice with ~10,000 -fold lower dose of bacteria than i.v. infection.

It remained unclear whether i.d. infection could also be used to model dissemination from the skin to internal organs. We therefore evaluated bacterial burdens in spleens and livers of WT and *Ifnar1*$^{-/-}$;*Ifngr1*$^{-/-}$ mice at 5 d.p.i. by measuring *R. parkeri* plaque-forming units (p.f.u.). Bacteria were not recoverable from spleens and livers of intradermally infected WT mice, suggesting that they did not disseminate from the skin to internal organs in high numbers (*Figure 2e*). In contrast, bacteria were recovered from spleens and livers of intradermally infected *Ifnar1*$^{-/-}$;*Ifngr1*$^{-/-}$ mice at 5 d.p.i. (*Figure 2e*). This demonstrates that i.d. infection of *Ifnar1*$^{-/-}$;*Ifngr1*$^{-/-}$ mice with *R. parkeri* causes systemic infection and can be used as a model for dissemination from the skin to internal organs.

## Histology and immunohistochemistry reveal necrosis, inflammation, and bacteria within neutrophils and macrophages

To examine whether lesions in *Ifnar1*$^{-/-}$;*Ifngr1*$^{-/-}$ mice had similar inflammation and necrosis as those observed in human eschars at the tissue and cellular level, we performed histologic evaluation to further characterize the tissue alterations of i.d.-infected *Ifnar1*$^{-/-}$;*Ifngr1*$^{-/-}$ mice and used anti-*Rickettsia* immunohistochemistry to identify the infected cell types in the skin. WT mice infected i.d. with $10^3$ *R. parkeri* exhibited no discernible inflammation in the skin at 4, 7, or 12 d.p.i. (*Figure 3a*). In contrast, at 4 d.p.i., *Ifnar1*$^{-/-}$;*Ifngr1*$^{-/-}$ mice infected with $10^3$ *R. parkeri* exhibited mild inflammation characterized by dermal inflammatory foci composed of neutrophils and macrophages, with high numbers of intralesional rickettsiae (*Figure 3b*). By 7 d.p.i., the inflammation throughout the skin was severe, with large coalescing foci of predominantly neutrophils and fewer macrophages within areas of fibrosis, associated with abundant rickettsiae. The epidermis had multifocal coagulative necrosis associated with

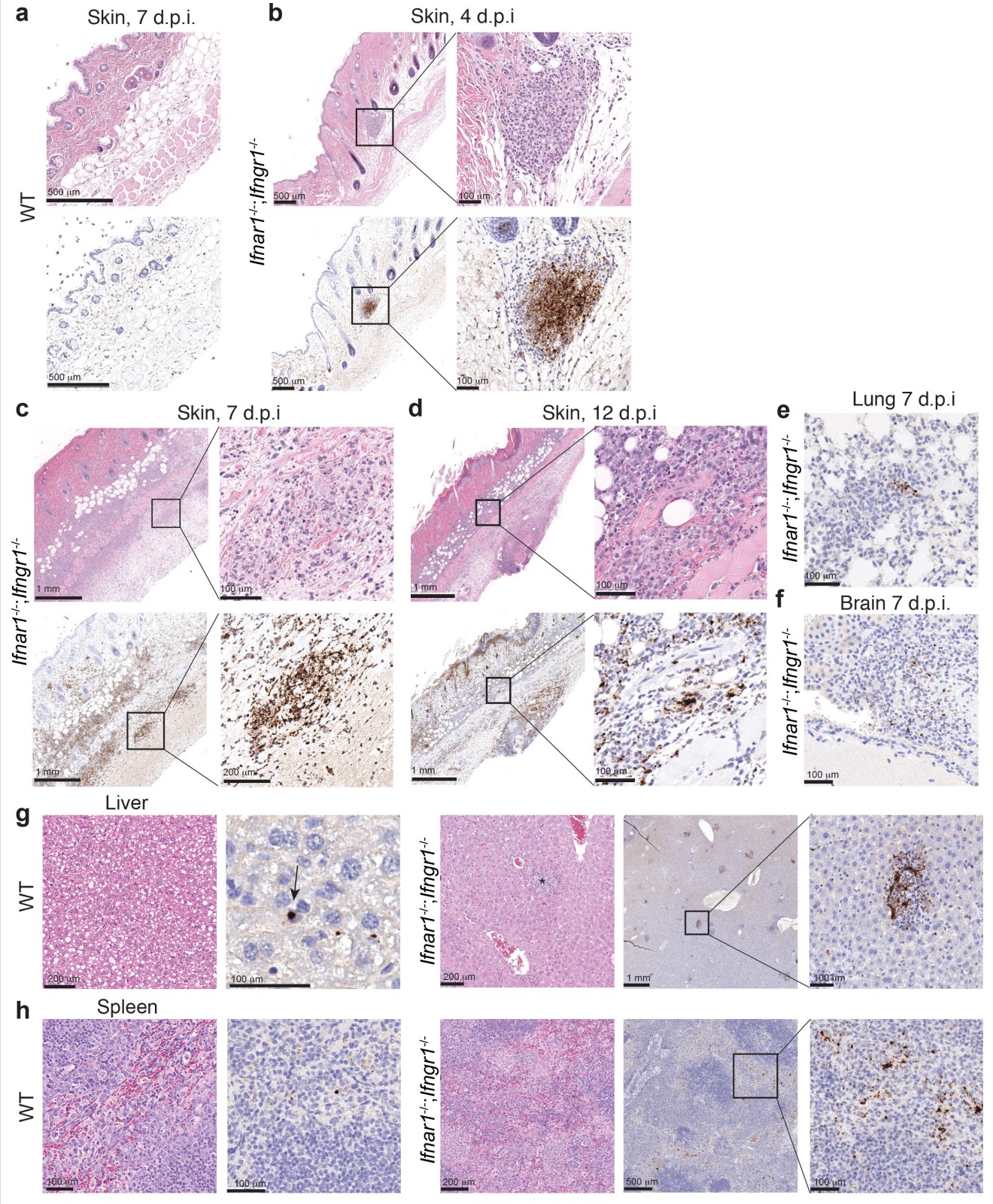

**Figure 3.** Pathology of infected *Ifnar1*⁻ᐟ⁻;*Ifngr1*⁻ᐟ⁻ mice reveals necrosis, inflammation, and rickettsial staining of macrophage and neutrophils. (**a–d**) Representative images of the skin from WT and *Ifnar1*⁻ᐟ⁻;*Ifngr1*⁻ᐟ⁻ mice, infected i.d. with 10³ *R. parkeri*. Top row panels are skin stained with hematoxylin and eosin. Bottom row panels are a nearby Z plane, stained with an anti-*Rickettsia* antibody (brown). (**e**) Representative image of lungs from *Ifnar1*⁻ᐟ⁻;*Ifngr1*⁻ᐟ⁻ mice, infected i.d. with 10³ *R. parkeri* at 7 d.p.i. (**f**) Representative image of brains (choroid plexus) from *Ifnar1*⁻ᐟ⁻;*Ifngr1*⁻ᐟ⁻ mice, infected i.d. with

*Figure 3 continued on next page*

Figure 3 continued

$10^6$ *R. parkeri* at 7 d.p.i. (**g, h**) Representative images of livers and spleens from WT and *Ifnar1*$^{-/-}$*;Ifngr1*$^{-/-}$ mice, infected i.v. with $10^7$ *R. parkeri* at 7 d.p.i. The arrow in (**g**) for WT mice indicates rickettsial staining within a Kupffer cell. The asterisk for *Ifnar1*$^{-/-}$*;Ifngr1*$^{-/-}$ indicates a granuloma. Scale bars are indicated in each panel. Data are representative of two independent replicates.

fibrin thrombosis of the underlying dermal capillaries (*Figure 3c*). The inflammation was still severe and associated with large numbers of intralesional rickettsiae at 12 d.p.i., accompanied by extensive coagulative necrosis of the skin (*Figure 3d*). Together, this indicates that *R. parkeri* causes extensive necrosis and inflammatory cell infiltration of the skin of i.d.-infected *Ifnar1*$^{-/-}$*;Ifngr1*$^{-/-}$ mice, with primary staining within neutrophils and macrophages.

We also examined internal organ tissues of mice infected i.d. by performing immunohistochemistry of brain and lung at 7 d.p.i., as infection of these organs may result in lethality. In the lungs of *Ifnar1*$^{-/-}$*;Ifngr1*$^{-/-}$ mice, inflammatory foci were scattered throughout the parenchyma, with several cells morphologically compatible with neutrophils containing rickettsiae (*Figure 3e*). In one tissue section, rickettsiae were also evident in some cells lining small vessels and therefore interpreted as endothelial cells. In the brain, rickettsial immunostaining was not observed after i.d. infection with $10^3$ *R. parkeri,* but infection with a higher dose ($10^6$) resulted in low number of rickettsial bacteria in the leptomeninges and the choroid plexus, in regions moderately infiltrated by neutrophils and macrophages (*Figure 3f*). These data demonstrate that *R. parkeri* disseminates from the skin to internal organs, where they are again primarily found in macrophages and neutrophils.

Finally, we used histology and immunohistochemistry after i.v. infection to analyze spleens and livers, as *R. parkeri* is abundant and easily recoverable from these organs after i.v. infection (*Burke et al., 2020*; *Engström et al., 2019*). In WT mice, we observed little to no inflammation or lesions. In contrast, both organs of *Ifnar1*$^{-/-}$*;Ifngr1*$^{-/-}$ mice had fibrinoid vascular wall degeneration, endothelial hypertrophy, fibrin thrombi in medium caliber vessels, and marked inflammation that was composed predominantly of macrophages. In WT mice, rickettsiae were infrequent and were found in macrophages in the red pulp of the spleen and scarcely throughout the liver in Kupffer cells (*Figure 3g,h*; *Bonell et al., 2017*). In contrast, in *Ifnar1*$^{-/-}$*;Ifngr1*$^{-/-}$ mice, rickettsiae were abundant in the splenic red pulp, primarily infecting histiocytes/macrophages (*Figure 3g*). In the liver of *Ifnar1*$^{-/-}$*;Ifngr1*$^{-/-}$ mice, rickettsiae were abundant in macrophages within the granulomas (*Figure 3h*). We did not observe any immunostaining for *Rickettsia* in endothelial cells or vessels in the spleen or liver after i.v. infection. Together, these findings reveal that macrophages are the primary cell type affected by *R. parkeri* in the spleen and liver after i.v. infection.

## *Ifnar1*$^{-/-}$*;Ifngr1*$^{-/-}$ mice do not succumb to intradermal infection with *Sca2* mutant *R. parkeri*

We next examined whether *Ifnar1*$^{-/-}$*;Ifngr1*$^{-/-}$ mice could be used to characterize *R. parkeri* virulence factors. Sca2 is a surface protein that mediates actin-based motility in rickettsial pathogens; however, its contribution to virulence in vivo remains unclear. We examined if i.v. and i.d. infections of WT and *Ifnar1*$^{-/-}$*;Ifngr1*$^{-/-}$ mice could reveal a pathogenic role for *R. parkeri* Sca2. Upon i.v. infection with $5 \times 10^6$ bacteria (*Figure 4*) or $10^7$ bacteria (*Figure 4b*), we observed that *sca2*::Tn mutant *R. parkeri* caused reduced lethality compared to WT bacteria. Similarly, i.d. infection with *sca2*::Tn mutant bacteria elicited significantly less lethality (*Figure 4c*) and weight loss (*Figure 4d*) as compared to WT bacteria and no severe temperature loss (*Figure 4—figure supplement 1a*). Although we sought to evaluate infection using a *sca2* complement strain of *R. parkeri,* our attempts to generate such a strain were unsuccessful. As an alternative strategy, we examined whether the transposon insertion itself had an effect on *R. parkeri* survival in vivo. We evaluated infection of an *R. parkeri* strain that harbors a transposon insertion in *MC1_RS08740* (previously annotated as *MC1_05535*), which has no known role in virulence (*Engström et al., 2019*). I.v. infection with *MC1_RS08740*::Tn *R. parkeri* caused lethality to a similar degree as WT *R. parkeri* (*Figure 4a*), demonstrating that the transposon likely does not significantly impact *R. parkeri* fitness in vivo. Together, these findings suggest that the actin-based motility factor Sca2 is required for causing lethal disease in *Ifnar1*$^{-/-}$*;Ifngr1*$^{-/-}$ mice.

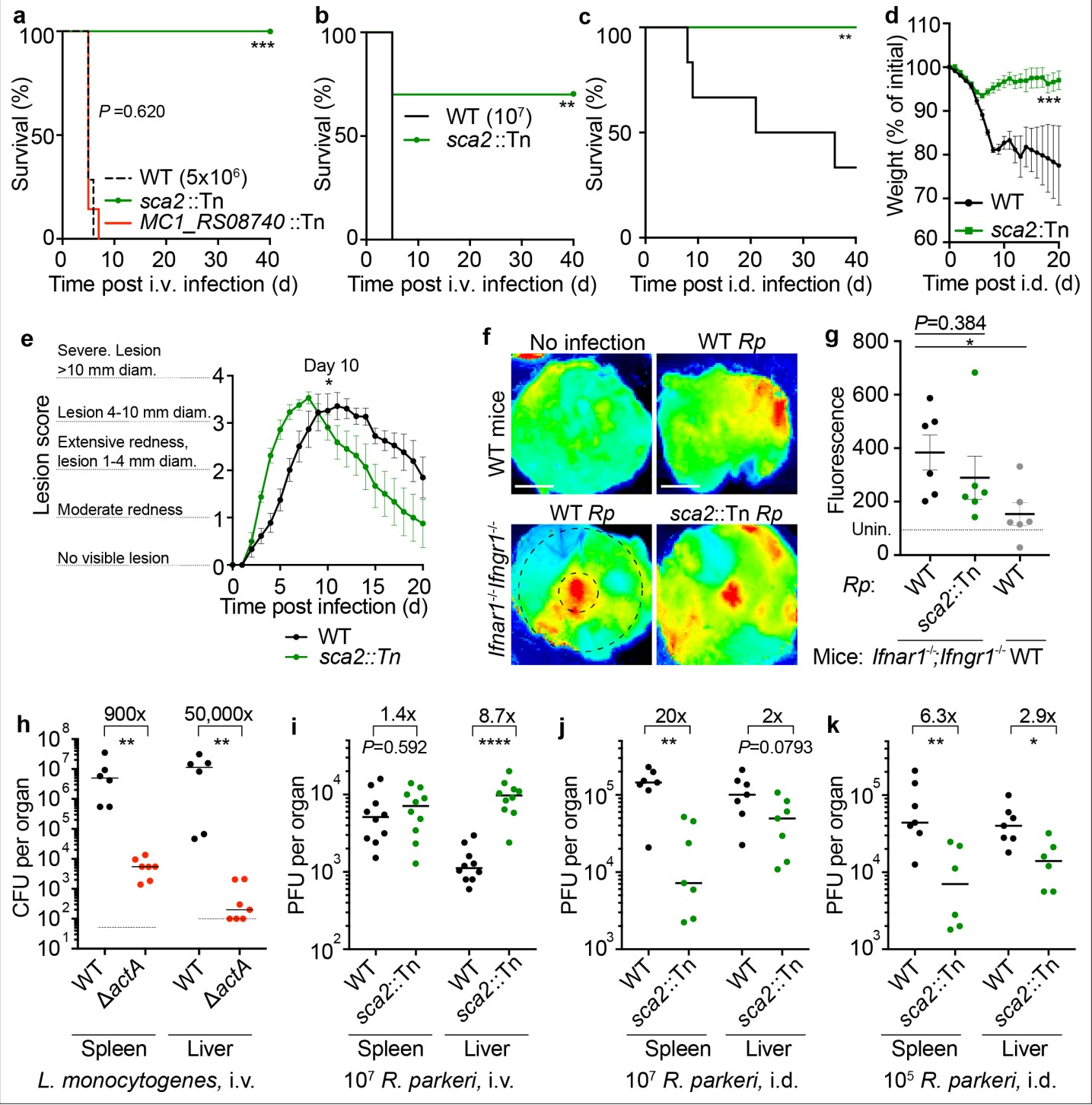

**Figure 4.** *R. parkeri* Sca2 contributes to dissemination from skin to spleens and livers. (**a**) Survival of *Ifnar1⁻ᐟ⁻;Ifngr1⁻ᐟ⁻* mice upon i.v. infection with 5 × 10⁶ PFU of *R. parkeri*. n = 7 (WT), 10 (*sca2*::Tn), and 7 (*MC1_RS08740*::Tn *R. parkeri*) individual mice. Data are the combination of two independent experiments. (**b**) Survival of *Ifnar1⁻ᐟ⁻;Ifngr1⁻ᐟ⁻* mice upon i.v. infection with 10⁷ *R. parkeri*. n = 7 (WT) and 10 (*sca2*::Tn) individual mice. Data are the combination of two independent experiments. (**c**) Survival of *Ifnar1⁻ᐟ⁻;Ifngr1⁻ᐟ⁻* mice upon i.d. infection with 10⁷ PFU of *R. parkeri*. *n* = 6 (WT) and 8 (*sca2*::Tn) individual mice. Data are the combination of two independent experiments. (**d**) Weight changes of *Ifnar1⁻ᐟ⁻;Ifngr1⁻ᐟ⁻* mice upon i.d. infection with 10⁷ PFU of *R. parkeri*. n = 6 (WT) and 8 (*sca2*::Tn) individual mice. Data are the combination of two independent experiments. (**e**) Analysis of gross skin pathology after i.d. infection. *Ifnar1⁻ᐟ⁻;Ifngr1⁻ᐟ⁻* mice were infected with 10⁷ PFU of the indicated strains of *R. parkeri* and monitored over time. n = 9 (WT) and 8 (*sca2*::Tn) individual mice. Data are the combination of two independent experiments. (**f**) Representative images of fluorescence in mouse skin after i.d. infection with 10⁶ *R. parkeri* and delivery of a fluorescent dextran, at 5 d.p.i. Scale bars, 1 cm. The larger black dashed circle represents

*Figure 4 continued on next page*

*Figure 4 continued*

the area that was measured for fluorescence for each sample, as indicated in (**g**) (~80,000 pixels). The smaller, black-dashed circle represents of the injection site area that was measured for fluorescence for each sample, as indicated in *Figure 4—figure supplement 1* (~7800 pixels). (**g**) Quantification of fluorescence in mouse skin after i.d. infection. Mice were infected with $10^7$ *R. parkeri*, and 150 μl fluorescent dextran was intravenously delivered at 5 d.p.i. Skin was harvested 2 hr later, and fluorescence was measured using a fluorescence imager. Data indicate measurements of larger areas of skin, as indicated in (**f**) by the larger black circle. n = 6 (WT *R. parkeri*) and n = 6 (*sca2*::Tn *R. parkeri*) individual *Ifnar1$^{-/-}$;Ifngr1$^{-/-}$* mice; n = 6 (WT *R. parkeri*) individual WT mice. For each experiment, the average of uninfected samples was normalized to 100; each sample was divided by the average for uninfected mice and multiplied by 100; the dotted horizontal line indicates 100 arbitrary units, corresponding to uninfected (unin.) mice. Data are the combination of two independent experiments. (**h**) Quantification of *Listeria monocytogenes* abundance in organs of WT C57BL/6 J mice upon i.v. infection with $10^4$ bacteria, at 72 h.p.i. Data are the combination of two independent experiments. n = 6 (WT), n = 7 (Δ*actA*) individual mice. (**i**) Quantification of *R. parkeri* abundance in spleens and livers of WT C57BL/6 J mice upon i.v. infection, at 72 h.p.i. Data are the combination of two independent experiments. n = 10 (WT) and 10 (*Sca2*::Tn) individual mice. (**j**) Quantification of *R. parkeri* abundance in organs upon i.d. infection with $10^7$ PFU of *R. parkeri*. n = 7 (WT) and 7 (*sca2*::Tn) individual mice. Data are the combination of two independent experiments. Data for WT *R. parkeri* in *Ifnar1$^{-/-}$;Ifngr1$^{-/-}$* mice are the same as in *Figure 2e*. (**k**) Quantification of *R. parkeri* abundance in organs upon i.d. infection with $10^5$ PFU of *R. parkeri*. n = 7 (WT) and 6 (*sca2*::Tn). Data are the combination of two independent experiments. Solid horizontal bars in (**g**) indicate means; solid horizontal bars in (**h–k**) indicate medians; error bars indicate SEM. Statistical analyses for survival in (**a–c**) used a log-rank (Mantel–Cox) test. Statistical analysis in (**d**) used a two-way ANOVA at t = 20. Statistical analysis in (**e**) used a two-way ANOVA from 0 to 10 d.p.i. Statistical analyses in (**g**) used a two-tailed Student's T test. Statistical analyses in (**h–k**) used a two-tailed Mann–Whitney U test. The fold change in (**h–k**) indicates differences of medians. *p<0.05; **p<0.01; ***p<0.001; ****p<0.0001.

The online version of this article includes the following source data and figure supplement(s) for figure 4:

**Source data 1.** *R. parkeri* Sca2 contributes to dissemination from skin to spleens and livers.

**Figure supplement 1.** Intradermal infection of *Ifnar1$^{-/-}$;Ifngr1$^{-/-}$* mice with *sca2*::Tn *R. parkeri* causes less severe temperature loss as compared to WT bacteria.

**Figure supplement 2.** WT and *sca2*::Tn *R. parkeri* elicit similar amounts of vascular damage in skin upon i.d. infection of *Ifnar1$^{-/-}$;Ifngr1$^{-/-}$* mice.

**Figure supplement 3.** Sca2 does not significantly enhance *R. parkeri* avoidance of antibacterial innate immune responses in vitro.

## *Ifnar1$^{-/-}$;Ifngr1$^{-/-}$* mice exhibit similar skin lesion formation and vascular damage upon i.d. infection with WT and *sca2*::Tn *R. parkeri*

We next examined whether Sca2 facilitates *R. parkeri* dissemination throughout the skin and whether Sca2 is required for lesion formation. Unexpectedly, upon i.d. inoculation, *Ifnar1$^{-/-}$;Ifngr1$^{-/-}$* mice infected with *sca2*::Tn mutant bacteria developed skin lesions that were of similar severity to lesions caused by WT *R. parkeri*; however, the lesions elicited by *sca2* mutant bacteria appeared significantly earlier than lesions caused by WT bacteria (*Figure 4e*). Further examinations will be required to better evaluate this observation; however, it may suggest that actin-based motility enables *R. parkeri* to avoid a rapid onset of inflammation in the skin. To evaluate *R. parkeri* dissemination within the skin, we used a fluorescence-based assay that measures vascular damage as a proxy for pathogen dissemination (*Glasner et al., 2017*). Mice were intradermally infected with WT and *sca2*::Tn *R. parkeri*. At 5 d.p.i., fluorescent dextran was intravenously delivered, and fluorescence was measured at the infection site (*Figure 4f*, representative small black circle) and in the surrounding area (*Figure 4f*, representative large black circle). No significant differences were observed when comparing WT and *sca2*::Tn *R. parkeri* infections in *Ifnar1$^{-/-}$;Ifngr1$^{-/-}$* mice using an infectious dose of $10^7$ *R. parkeri* in the larger surrounding area (*Figure 4g*) or at the site of infection (*Figure 4—figure supplement 2a*). Similar results were observed upon infection with $10^6$ or $10^5$ bacteria (*Figure 4—figure supplement 2b,c*). However, significantly more fluorescence was observed in the skin of infected *Ifnar1$^{-/-}$;Ifngr1$^{-/-}$* mice as compared to WT mice (*Figure 4g*), demonstrating that interferons protect against increased vascular permeability during *R. parkeri* infection. Fluorescence was also measured in spleens and livers; however, no differences between control or experiment groups was observed, suggesting that this assay as described is most appropriate for the skin. Together, the gross pathological analysis and fluorescence-based assay suggest that Sca2 likely does not significantly enhance *R. parkeri* spread throughout the skin during i.d. infection of *Ifnar1$^{-/-}$;Ifngr1$^{-/-}$* mice.

## *R. parkeri* Sca2 promotes dissemination from the skin to spleens and livers

Among the factors that mediate actin-based motility, the *Listeria monocytogenes* actin-based motility factor ActA is one of the best understood. ActA enables *L. monocytogenes* to spread from cell to

cell (*Choe and Welch, 2016*; *Lamason and Welch, 2017*), escape antimicrobial autophagy (*Cheng et al., 2018*; *Mitchell et al., 2018*; *Yoshikawa et al., 2009b*; *Yoshikawa et al., 2009a*), proliferate in mouse organs after i.v. infection (*Auerbuch et al., 2001*; *Le Monnier et al., 2007*), and cause lethal disease in mice (*Goossens et al., 1992*; *Brundage et al., 1993*). We initially hypothesized that *R. parkeri* Sca2 plays a similar pathogenic role in vivo to ActA, which we found is required for bacterial survival in spleens and livers upon i.v. delivery (*Figure 4h*), in agreement with previous experiments (*Auerbuch et al., 2001*; *Le Monnier et al., 2007*). However, when we examined bacterial burdens upon i.v. infection of WT mice with *R. parkeri*, similar amounts of WT and *sca2*::Tn bacteria were recovered in spleens (*Figure 4i*). We were also surprised to find that significantly more *sca2*::Tn than WT *R. parkeri* were recovered in livers (*Figure 4i*). The explanation for higher *sca2*::Tn burdens in livers remains unclear. Nevertheless, these data reveal that Sca2 is likely not essential for *R. parkeri* survival in blood, invasion of host cells, or intracellular survival in spleens and livers.

We next evaluated the role for Sca2 in *R. parkeri* dissemination by measuring p.f.u. in spleens and livers following i.d. infection of *Ifnar1*[-/-];*Ifngr1*[-/-] mice. After i.d. infection, *sca2*::Tn mutant bacteria were ~20 -fold reduced in their abundance in spleens and ~2 -fold reduced in their abundance in livers as compared to WT *R. parkeri* (*Figure 4j*). Similar results were seen upon i.d. infection with lower doses of *sca2*::Tn and WT bacteria (*Figure 4k*). Together, these results suggest that Sca2 is required for *R. parkeri* dissemination from the skin to internal organs.

### *R. parkeri* actin-based motility does not contribute to avoiding innate immunity in vitro

Sca2-mediated actin-based motility is required for efficient plaque formation and cell to cell spread by *R. parkeri* in vitro (*Reed et al., 2014*; *Kleba et al., 2010*). However, it remains unclear if Sca2 enables *R. parkeri* to escape detection or restriction by innate immunity. The actin-based motility factor ActA enables *L. monocytogenes* to avoid autophagy (*Yoshikawa et al., 2009b*; *Yoshikawa et al., 2009a*), and the antimicrobial guanylate binding proteins (GBPs) inhibit *Shigella flexneri* actin-based motility (*Piro et al., 2017*). We therefore sought to evaluate whether Sca2-mediated actin-based motility enables *R. parkeri* to evade innate immunity in vitro. We found that the *sca2*::Tn mutant grew similarly to WT bacteria in endothelial cells (*Figure 4—figure supplement 3a*), consistent with previous reports in epithelial cells (*Reed et al., 2014*; *Kleba et al., 2010*). We also examined whether Sca2 contributed to *R. parkeri* survival or growth in bone marrow-derived macrophages (BMDMs), which can restrict other *R. parkeri* mutants that grow normally in endothelial cells (*Engström et al., 2019*). However, no significant difference in bacterial survival was observed between WT and *sca2*::Tn bacteria in BMDMs in the presence or absence of IFN-β (*Figure 4—figure supplement 3b*). WT and *sca2* mutant *R. parkeri* also elicited similar amounts of host cell death (*Figure 4—figure supplement 3c*) and IFN-I production (*Figure 4—figure supplement 3d*). Moreover, we found that the anti-rickettsial factor GBP2 localized to the surface of *sca2*::Tn mutant *R. parkeri* at similar frequency as with WT bacteria in the presence or absence of IFN-β (*Figure 4—figure supplement 3e,f*). Together, these data suggest that Sca2 does not significantly enhance the ability of *R. parkeri* to evade interferon-stimulated anti-microbial genes or inflammasomes in vitro.

### *Ifnar1*[-/-];*Ifngr1*[-/-] mice exhibit less severe skin lesions upon infection with a highly attenuated *R. parkeri* mutant

Because *sca2* mutant *R. parkeri* showed no defect in eschar formation compared to WT, it remained unclear whether skin lesion formation in *Ifnar1*[-/-];*Ifngr1*[-/-] mice was influenced by bacterial virulence factors. We therefore investigated i.d. infection with *ompB*::Tn[STOP] *R. parkeri,* which harbors both a transposon and a stop codon in *ompB* but no other mutations as determined by whole genome sequencing (*Engström et al., 2019*). *ompB* mutant *R. parkeri* are severely attenuated in mice, as evaluated by measuring p.f.u. in organs of WT mice after i.v. infection (*Engström et al., 2019*) or by measuring lethality in *Ifnar1*[-/-];*Ifngr1*[-/-] mice after i.v. infection (*Burke et al., 2020*). In contrast with WT bacteria, i.d. infection of *Ifnar1*[-/-];*Ifngr1*[-/-] mice with *ompB*::Tn[STOP] *R. parkeri* caused no lethality (*Figure 5a*) or weight loss (*Figure 5b*). The *ompB*::Tn[STOP] mutant *R. parkeri* also caused significantly less severe skin lesions than WT bacteria (*Figure 5c*). These findings suggest that *Ifnar1*[-/-];*Ifngr1*[-/-] mice can be used as a model to identify bacterial genes important for eschar formation.

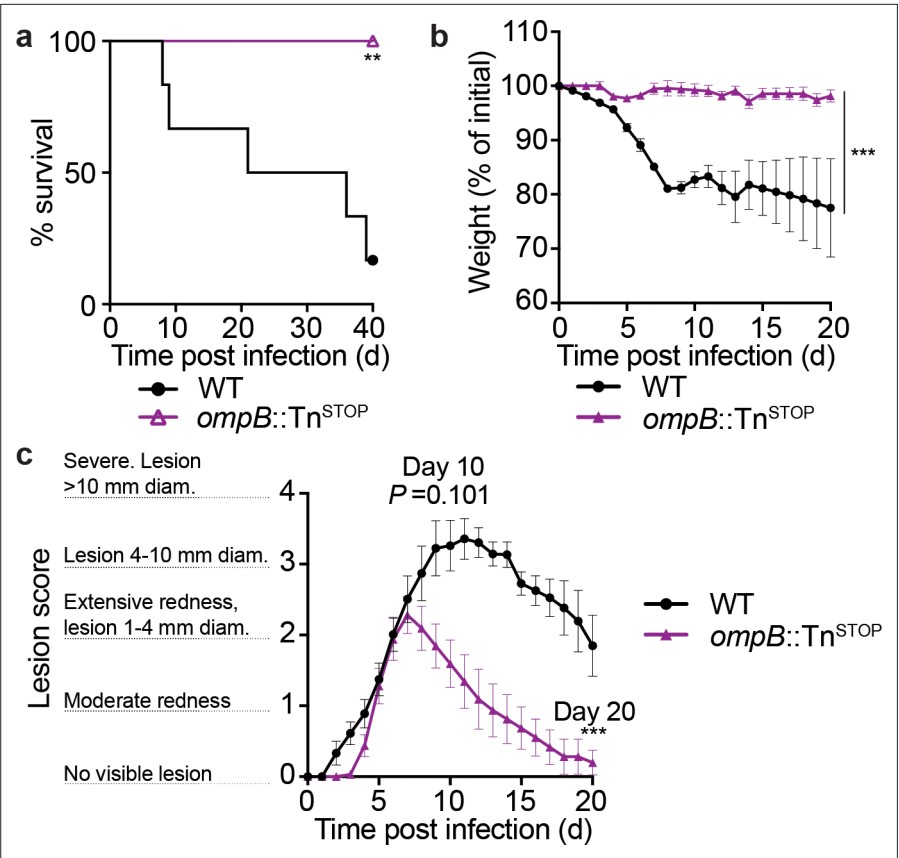

**Figure 5.** *ompB* mutant *R. parkeri* elicit no lethality and reduced skin lesion formation in *Ifnar1⁻/⁻;Ifngr1⁻/⁻* mice.
(**a**) Survival of *Ifnar1⁻/⁻;Ifngr1⁻/⁻* mice upon i.d. infection with $10^7$ PFU of *R. parkeri*. n = 6 (WT) and 8 (*ompB*::Tn^STOP^) individual mice. Data are the combination of two independent experiments. Data for WT are the same as in **Figure 4c**. (**b**) Weight changes of *Ifnar1⁻/⁻;Ifngr1⁻/⁻* mice upon i.d. infection with $10^7$ PFU of *R. parkeri*. n = 6 (WT) and 8 (*ompB*::Tn^STOP^) individual mice. Data are the combination of two independent experiments. Data for WT are the same as in **Figure 4d**. (**c**) Analysis of gross skin pathology after i.d. infection. *Ifnar1⁻/⁻;Ifngr1⁻/⁻* mice were infected with $10^7$ of the indicated strains of *R. parkeri* and monitored over time. n = 9 (WT) and 8 (*ompB*::Tn^STOP^) individual mice. Data are the combination of two independent experiments. Data for WT are the same as in **Figure 4e**. Error bars indicate SEM. Statistical analyses in (**a**) used a log-rank (Mantel–Cox) test. Statistical analysis in (**b**) used a two-way ANOVA from 0 to 20 d.p.i. Statistical analysis in (**c**) used a two-way ANOVA from 0 to 10 and 20 d.p.i.; **p<0.01; ***p<0.001.

The online version of this article includes the following source data for figure 5:

**Source data 1.** *ompB* mutant *R. parkeri* elicit no lethality and reduced skin lesion formation in *Ifnar1⁻/⁻;Ifngr1⁻/⁻* mice.

## Immunizing *Ifnar1⁻/⁻;Ifngr1⁻/⁻* mice with attenuated *R. parkeri* mutants protects against subsequent rechallenge

There is currently no available vaccine to protect against SFG *Rickettsia,* which can cause severe and lethal human disease (**Osterloh, 2017**; **Dantas-Torres, 2007**), and identifying mouse models that develop protective immunity to *R. parkeri* would aid investigations into identifying live attenuated vaccine candidates. We therefore examined whether immunization with attenuated *R. parkeri* mutants would protect against subsequent re-challenge with a lethal dose of WT bacteria. *Ifnar1⁻/⁻;Ifngr1⁻/⁻* mice were immunized i.v. with $5 \times 10^6$ *sca2*::Tn or *ompB*::Tn^STOP^ *R. parkeri* and 40 d later were re-challenged i.v. with $10^7$ WT *R. parkeri*, which is approximately 10 times a 50 % lethal dose (LD$_{50}$) (**Burke et al., 2020**). All mice immunized with *sca2* or *ompB* mutant *R. parkeri* survived, whereas all naïve mice succumbed to i.v. challenge by 6 d.p.i. (**Figure 6a**). Upon i.v. rechallenge, mice immunized with *ompB* and *sca2* mutants also did not lose significant weight (**Figure 6b**) or body temperature (**Figure 6c**). We next examined whether immunized mice were protected from i.d. rechallenge. *Ifnar1⁻/⁻;Ifngr1⁻/⁻* mice

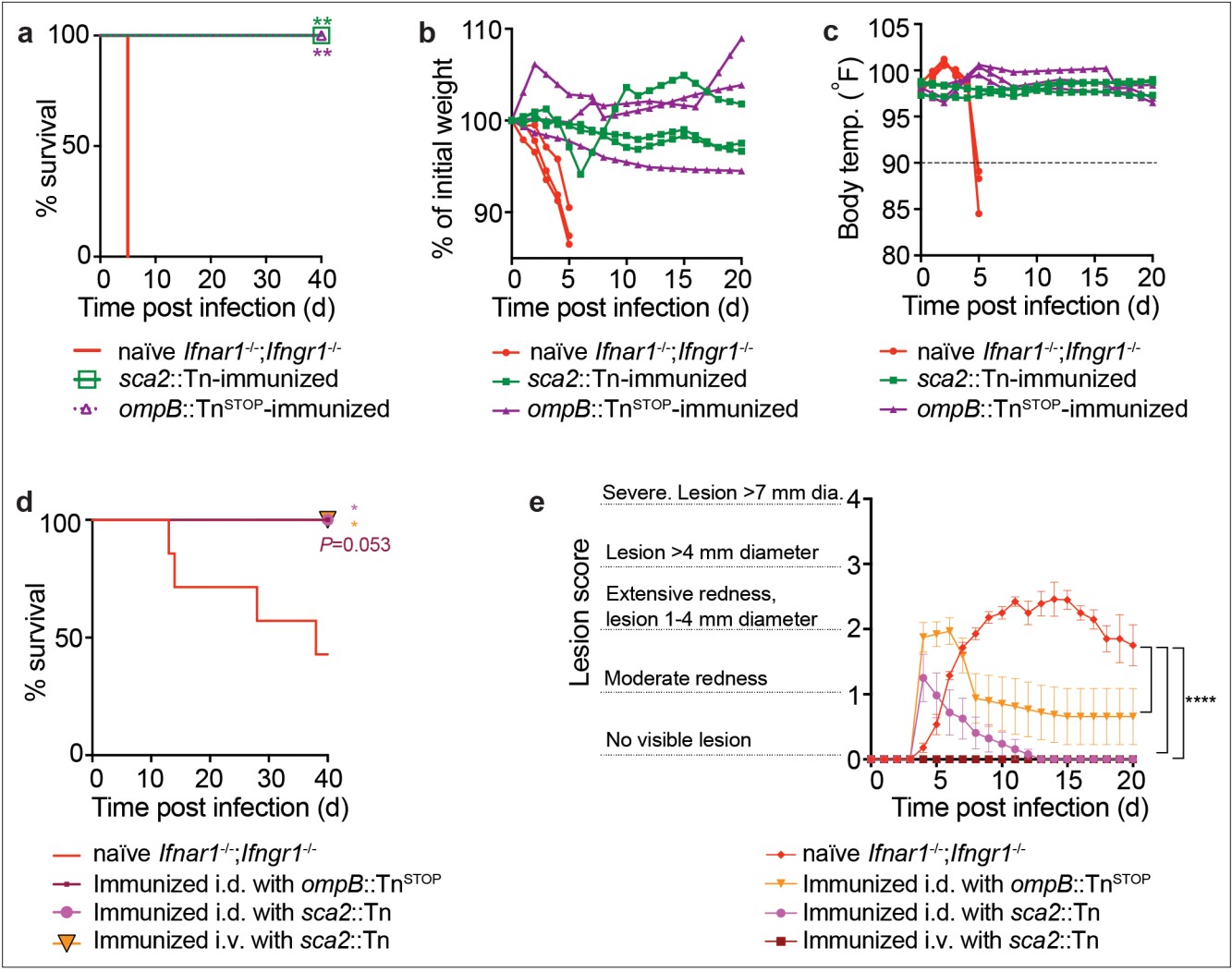

**Figure 6.** *ompB* and *sca2* mutant *R. parkeri* elicit immunity in *Ifnar1⁻/⁻ Ifngr1⁻/⁻* mice. (**a**) Survival of immunized and naïve *Ifnar1⁻/⁻;Ifngr1⁻/⁻* mice upon i.v. *R. parkeri* infection. Immunized mice were first infected with 5 × 10⁶ PFU of *sca2*::Tn or 10⁷ PFU of *ompB*:Tnˢᵀᴼᴾ *R. parkeri* and were re-challenged 40 d later with 10⁷ WT *R. parkeri*. n = 6 (naïve); n = 5 (*sca2*::Tn immunized); n = 5 (*ompB*::Tnˢᵀᴼᴾ immunized) individual mice. Data are the combination of two independent experiments. (**b**) Weight changes over time in mice infected i.d. with 10⁷ *R. parkeri*. Data are representative of two independent experiments. n = 3 (naïve); n = 3 (*sca2*::Tn immunized); n = 3 (*ompB*::Tnˢᵗᵒᵖ immunized) individual mice. Each line represents an individual mouse. (**c**) Temperature changes over time in mice infected i.d. with 10⁷ *R. parkeri*. Data are representative from two independent experiments. n = 3 (naïve); n = 3 (*sca2*::Tn immunized); n = 3 (*ompB*::Tnˢᵀᴼᴾ immunized) individual mice. Each line represents an individual mouse. (**d**) Survival of naïve or immunized *Ifnar1⁻/⁻;Ifngr1⁻/⁻* mice that were infected i.d. with 10⁵ *R. parkeri*. (**e**) Skin lesion pathology of naïve or immunized *Ifnar1⁻/⁻;Ifngr1⁻/⁻* mice that were infected i.d. with 10⁵ *R. parkeri*; n = 7 (naïve); n = 8 (*sca2*::Tn i.d. immunized); n = 5 (*sca2*::Tn i.v. immunized); n = 8 (*ompB*::Tnˢᵀᴼᴾ) individual mice. Statistical analyses in (**a, d**) used a log-rank (Mantel–Cox) test to compare each group of immunized mice to naïve mice; **p<0.01. Statistical analysis in (**e**) used a two-way ANOVA from 0 to 10 and 20 d.p.i.; **p<0.01; ***p<0.001.

The online version of this article includes the following source data for figure 6:

**Source data 1.** *ompB* and *sca2* mutant *R. parkeri* elicit immunity in *Ifnar1⁻/⁻ Ifngr1⁻/⁻* mice.

were immunized with *ompB* or *sca2* mutant *R. parkeri* by i.d. infection or with *sca2* mutant *R. parkeri* by i.v infection and rechallenged 40 d later by i.d. infection with 10⁵ WT bacteria. Immunized mice were protected from lethal disease (**Figure 6d**) and had less severe skin lesions than naïve mice (**Figure 6e**). These data indicate that attenuated *R. parkeri* mutants elicit robust protective immune responses, and that *Ifnar1⁻/⁻;Ifngr1⁻/⁻* mice may serve as tools to develop live attenuated *R. parkeri* vaccine candidates.

## Discussion

In this study, we find that IFN-I and IFN-γ redundantly protect inbred mice from eschar-associated rickettsiosis and disseminated disease by *R. parkeri*. Eschar formation is a hallmark clinical feature of human disease caused by *R. parkeri* (*Paddock et al., 2008*; *Paddock et al., 2004*), and thus these findings suggest that the striking difference between human and mouse susceptibilities to *R. parkeri* may be due to IFN signaling in the skin. Using this mouse model, we uncover a role for *R. parkeri* Sca2 in dissemination, for OmpB in skin lesion formation, and for both proteins in causing lethal disease. We further demonstrate that attenuated *R. parkeri* mutants elicit long-lasting immunity, revealing live attenuated vaccine candidates. Obligate cytosolic bacterial pathogens cause a variety of severe human diseases on six continents (*Bonell et al., 2017*; *Abdad et al., 2018*), and the animal model described here will facilitate future investigations into rickettsial virulence factors, the host response to infection, and the molecular determinants of human disease.

Our observation that i.d. infection of DKO $Ifnar1^{-/-};Ifngr1^{-/-}$ mice causes eschar formation highlights the critical importance of interferons in restricting *R. parkeri* in mice and may reveal a key molecular determinant of human disease. IFN-γ restricts virulent rickettsial species in vitro and in mice, including *R. prowazekii* (*Turco and Winkler, 1983*), *R. conorii* (*Feng et al., 1994*; *Li et al., 1987*; *Manor and Sarov, 1990*), and *R. australis* (*Walker et al., 2001*). Moreover, IFN-I has modest anti-rickettsial activity towards *R. prowazekii*, *R. conorii*, and *R. rickettsii* in cell lines in vitro (*Turco and Winkler, 1990*; *Colonne et al., 2011*; *Hanson, 1991*). However, the contribution made by IFN-I in restricting *Rickettsia* species besides *R. parkeri* in mice remains unknown (*Osterloh, 2017*). Importantly, the gross similarities we observe between human eschars (*Paddock et al., 2008*; *Herrick et al., 2016*) and skin lesions in $Ifnar1^{-/-};Ifngr1^{-/-}$ mice may indicate that the IFN response in humans is less well adapted to control *R. parkeri* than that in mice. Cytokine profiling and mRNA transcript analysis of human rickettsial infections reveals IFN-I and IFN-γ activation (*Cillari et al., 1996*; *Jia et al., 2020*; *de Sousa et al., 2007*); however, it remains unclear whether human interferon-stimulated genes (ISGs) are as protective as mouse ISGs. Future investigations into the ISGs that restrict *R. parkeri* in mouse versus human cells may improve our understanding of human susceptibility to SFG *Rickettsia*.

Human eschars resulting from *R. parkeri* infection are characterized by necrosis of the epidermis, vasculitis of small- to medium-sized dermal vessels, fibrin thrombi, infiltration by macrophages and neutrophils, and infection of macrophage/mononuclear cells (*Paddock et al., 2008*; *Herrick et al., 2016*). Eschars elicited by *R. parkeri* in non-human primates revealed similar histological findings and also bacteria within macrophages and neutrophils (*Banajee et al., 2015*). Similar histological findings were noted in guinea pigs after intraperitoneal infection with *R. parkeri*, as well as infection of mononuclear cells (*Paddock et al., 2017*). In $Ifnar1^{-/-};Ifngr1^{-/-}$ mice, we observed necrosis, vasculitis, fibrin thrombi, and rickettsial staining that coincided primarily with infiltrating macrophages and neutrophils. We note that *R. parkeri* primary targeting of macrophages and neutrophils differs from cell types infected by *R. rickettsii*, which primarily targets endothelial cells (*Walker and Ismail, 2008*), and therefore infection of $Ifnar1^{-/-};Ifngr1^{-/-}$ mice with *R. parkeri* is likely not an appropriate model for lethal Rocky Mountain spotted fever disease. One contrasting finding in eschars of $Ifnar1^{-/-};Ifngr1^{-/-}$ mice as compared to humans is the number of bacteria observed - we find that the pathogen is numerous throughout the skin of $Ifnar1^{-/-};Ifngr1^{-/-}$ mice, but their numbers are fewer in reported human eschar biopsies (*Paddock et al., 2008*; *Herrick et al., 2016*). This discrepancy may be due to differences in infection kinetics or in interferon-mediated restriction of bacteria. Nevertheless, our pathological findings of infected interferon-receptor deficient mice are similar in many ways to the previous findings in humans, primates, and mice. Our results suggest that interferon receptor-deficient mice may be useful tools for modeling eschar-associated rickettsiosis at the cellular level.

Investigating the IFN response in the skin may also lead to a better understanding of other arthropod-borne diseases. One example may be scrub typhus, caused by *Orientia tsutsugamushi* (*Rajapakse et al., 2017*), a prevalent but poorly understood tropical disease endemic to Southeast Asia (*Bonell et al., 2017*; *Kelly et al., 2015*; *Richards and Jiang, 2020*). *O. tsutsugamushi* elicits eschar formation in humans, but inbred mice do not recapitulate eschar formation during *O. tsutsugamushi* infection (*Sunyakumthorn et al., 2013*), similar to *R. parkeri*. A second example may be *Borrelia burgdorferi*, a tick-borne pathogen that causes a skin rash at the site of tick bite as a hallmark feature of Lyme disease (*Bratton et al., 2008*), the most prevalent tick-borne

disease in the United States (*Bratton et al., 2008*; *Schwartz et al., 2008*). Existing mouse models also do not recapitulate skin rash formation following *B. burgdorferi* infection (*Barthold et al., 1990*; *Wang et al., 2001*). Lastly, other rickettsial pathogens including *R. conorii, R. typhi,* and *R. akari* cause no or limited disease in WT C57BL/6 mice (*Osterloh, 2017*; *Anderson and Osterman, 1980*; *Eisemann et al., 1984*; *Osterloh et al., 2016*) and it remains unclear if interferons redundantly protect mice from these pathogens. Further investigations into how IFNs protect the skin from arthropod-borne pathogens may reveal critical aspects of the innate immune response to zoonotic disease.

Our study further highlights the utility of mouse models that mimic natural routes of infection. Our observation that i.d. infection can cause lethal disease with as few as 10 bacteria, ~ 10,000 fewer bacteria than i.v. infection (*Burke et al., 2020*), suggests that *R. parkeri* may be highly adapted to reside in the skin. However, this model could be further improved by investigating the role for tick vector components in pathogenesis. Saliva from ticks, mosquitos, and sand flies enhances pathogenesis of arthropod-borne bacterial, viral, and parasitic pathogens (*Pingen et al., 2017*; *Lestinova et al., 2017*; *Šimo et al., 2017*), and non-human primates inoculated with *R. parkeri* exhibit altered inflammatory responses when administered after tick-bite (*Banajee et al., 2015*). This may suggest a potential role for tick vector components such as tick saliva in *R. parkeri* pathogenesis. Developing improved murine infection models that mimic the natural route of infection, including with tick saliva or the tick vector, is critical to better understand the virulence and transmission of tick-borne pathogens.

Many *Rickettsia* species, as well as many facultative cytosolic pathogens including *L. monocytogenes*, undergo actin-based motility to spread from cell to cell. For *L. monocytogenes*, the actin-based motility factor ActA enables the pathogen to survive in vivo, as *actA* mutant bacteria are over 1,000-fold attenuated by measuring lethality (*Goossens et al., 1992*; *Brundage et al., 1993*) and by enumerating bacteria in spleens and livers of mice after i.v. infection (*Auerbuch et al., 2001*; *Le Monnier et al., 2007*). However, the pathogenic role for actin-based motility in the Rickettsiae has remained unclear. We find that Sca2 is not required for intracellular survival in organs upon i.v. infection of *Ifnar1-/-;Ifngr1-/-* mice, but rather, is required for dissemination from skin to internal organs and lethality upon i.d. infection. Consistent with an important role for Sca2 in pathogenesis, a previous study reported that i.v. infection of guinea pigs with *sca2* mutant *R. rickettsii* did not elicit fever (*Kleba et al., 2010*). Our results suggest that Sca2-mediated actin-based motility by *Rickettsia* may facilitate dissemination in host reservoirs, although we cannot rule out other roles for Sca2 that do not involve actin assembly. *R. prowazekii* and *R. typhi*, which cause severe human disease, encode a fragmented *sca2* gene (*Ngwamidiba et al., 2005*), and undergo no or dramatically reduced frequency of actin-based motility, respectively (*Teysseire et al., 1992*; *Heinzen et al., 1993*). Although it remains unclear why some *Rickettsia* species lost the ability to undergo actin-based motility, Sca2 is dispensable for *R. parkeri* dissemination in the tick vector (*Harris et al., 2018*), suggesting that actin-based motility may play a specific role in dissemination within mammalian hosts.

We find that *sca2* or *ompB* mutant *R. parkeri* elicit a robust protective immune response in *Ifnar1-/-;Ifngr1-/-* mice. These findings complement previous observations that *sca2* mutant *R. rickettsii* elicits antibody responses in guinea pigs (*Kleba et al., 2010*), and expands upon these findings by demonstrating protection from rechallenge and revealing additional vaccine candidates. There are currently limited vaccine candidates that protect against rickettsial disease (*Osterloh, 2017*). Identifying new vaccine candidates may reveal avenues to protect against tick-borne infections and aerosolized *Rickettsia*, which are extremely virulent and potential bioterrorism agents (*Walker, 2009*), as well as against Brill-Zinsser disease, caused by latent *R. prowazekii* (*Osterloh, 2017*). Future studies exploring whether attenuated *R. parkeri* mutants provide immunity against other *Rickettsia* species are warranted to better define the mechanisms of protection. These findings on immunity may also help develop *R. parkeri* as an antigen delivery platform. *R. parkeri* resides directly in the host cytosol for days and could potentially be engineered to secrete foreign antigens for presentation by major histocompatibility complex I. In summary, the mouse model described here will facilitate future investigations into numerous aspects of *R. parkeri* infection, including actin-based motility and immunity, and may serve as model for other arthropod-borne pathogens.

# Materials and methods

## Key resources table

| Reagent type (species) or resource | Designation | Source or reference | Identifiers | Additional information |
|---|---|---|---|---|
| Biological sample (*Mus musculus*) | WT C57Bl/6J | Jackson labs | Stock #: 000664 | |
| Biological sample (*Mus musculus*) | DKO *Ifnar1$^{-/-}$;Ifngr1$^{-/-}$* | Jackson labs | Stock #: 029098 | |
| Biological sample (*Mus musculus*) | B6.129S2-*Ifnar1$^{tm1Agt}$*/Mmjax; aka *Ifnar1$^{-/-}$* | Jackson labs | Stock #: 32045-JAX | RRID:MMRRC_032045-JAX |
| Biological sample (*Mus musculus*) | B6.129S7-*Ifngr1$^{tm1Agt}$*/J; aka *Ifngr1$^{-/-}$* | Jackson labs | Stock #: 003288 | |
| Antibody | I7205 anti-*Rickettsia* (Rabbit polyclonal) | Dr. Ted Hackstadt | | IHC (1:1000); IF (1:1000) |
| Biological sample (*Rickettsia parkeri*) | WT *R. parkeri* strain Portsmouth | Originally from Dr. Chris Paddock (CDC) | | |
| Biological sample (*Rickettsia parkeri*) | *Sca2*::Tn *R. parkeri* | Welch lab (UC Berkeley) PMID:24361066 | | |
| Biological sample (*Rickettsia parkeri*) | *OmpB*::Tn$^{STOP}$*R. parkeri* | Welch lab (UC Berkeley) PMID:31611642 | | |

## Bacterial preparations

*R. parkeri* strain Portsmouth was originally obtained from Dr. Christopher Paddock (Centers for Disease Control and Prevention). To amplify *R. parkeri*, confluent monolayers of female African green monkey kidney epithelial Vero cells (obtained from UC Berkeley Cell Culture Facility, tested for myco-plasma contamination, and authenticated by mass spectrometry) were infected with $5 \times 10^6$ *R. parkeri* per T175 flask. Vero cells were grown in DMEM (Gibco 11965–092) containing 4.5 g l$^{-1}$ glucose and 2 % fetal bovine serum (FBS; GemCell). Infected cells were scraped and collected at five or 6 d.p.i. when ~90 % of cells were rounded due to infection. Scraped cells were then centrifuged at 12,000 g for 20 min at 4 °C. Pelleted cells were resuspended in K-36 buffer (0.05 M KH$_2$PO$_4$, 0.05 M K$_2$HPO$_4$, 100 mM KCl, 15 mM NaCl, pH 7) and dounced for ~40 strokes at 4 °C. The suspension was then centrifuged at 200 g for 5 min at 4 °C to pellet host cell debris. Supernatant containing *R. parkeri* was overlaid on a 30 % MD-76R (Merry X-Ray) gradient solution in ultracentrifuge tubes (Beckman/Coulter Cat 344058). Gradients were centrifuged at 18,000 r.p.m. in an SW-28 ultracentrifuge swinging bucket rotor (Beckman/Coulter) for 20 min at 4 °C. These '30 % prep' bacterial pellets were resuspended in brain heart infusion (BHI) media (BD, 237500), aliquoted, and stored at −80 °C. Titers were deter-mined by plaque assays by serially diluting the *R. parkeri* in six-well plates containing confluent Vero cells. Plates were spun for 5 min at 300 g in an Eppendorf 5810 R centrifuge and at 24 hr post infection (h.p.i.); the media from each well was aspirated, and the wells were overlaid with 4 ml/well DMEM with 5 % FBS and 0.7 % agarose (Invitrogen, 16500–500). At 6 d.p.i., an overlay of 0.7 % agarose in DMEM containing 2.5 % neutral red (Sigma, N6264) was added. Plaques were then counted 24 hr later. For infections with *ompB* mutant bacteria, the *ompB$^{STOP}$*::Tn mutant was used, which contains a transposon and an upstream stop codon in *ompB*, as previously described (*Engström et al., 2019*).

## Deriving bone marrow macrophages

For obtaining bone marrow, male or female mice were euthanized, and femurs, tibias, and fibulas were excised. Bones were sterilized with 70 % ethanol and washed with BMDM media (20 % FBS (HyClone), 0.1 % β-mercaptoethanol, 1 % sodium pyruvate, 10 % conditioned supernatant from 3T3 fibroblasts, in DMEM (Gibco) with 4.5 g l$^{-1}$ glucose and 100 µg/ml streptomycin and 100 U/ml penicillin), and ground with a mortar and pestle. Bone homogenate was passed through a 70 µm nylon cell strainer (Thermo Fisher Scientific, 08-771-2) for particulate removal. Filtrates were then centrifuged at 290 g in an Eppendorf 5810 R centrifuge for 8 min, supernatant was aspirated, and the pellet was resuspended in BMDM media. Cells were plated in 30 ml BMDM media in non-TC-treated 15 cm petri dishes at a ratio of 10 dishes per two femurs/tibias and incubated at 37 °C. An additional 30 ml of BMDM media was added 3 d later. At 7 d the media was aspirated, 15 ml cold PBS (Gibco, 10010–023) was added, and cells were incubated at 4 °C with for 10 min. BMDMs were scraped from the plate, collected in a

50 ml conical tube, and centrifuged at 290 g for 5 min. PBS was aspirated, and cells were resuspended in BMDM media with 30 % FBS and 10 % DMSO at $10^7$ cells/ml. One milliliter of aliquots was stored at –80 °C for 24 hr in Styrofoam boxes and then moved to long-term storage in liquid nitrogen.

## Infections in vitro

HMEC-1 cells (obtained from the UC Berkeley Cell Culture Facility and authenticated by short-tandem-repeat analysis) were passaged two to three times weekly and grown at 37 °C with 5 % $CO_2$ in DMEM containing 10 mM L-glutamine (Sigma, M8537), supplemented with 10 % heat-inactivated FBS (HyClone), 1 µg/ml hydrocortisone (Spectrum Chemical, CO137), 10 ng/ml epidermal growth factor (Thermo Fisher Scientific, CB40001; Corning cat. no. 354001), and 1.18 mg/ml sodium bicarbonate. HMEC media was prepared every 1–2 months and aliquoted and stored at 4 °C. To prepare HMEC-1 cells for infection, cells were treated with 0.25 % trypsin-EDTA (Thermo Fisher Scientific); the number of cells was counted using a hemocytometer (Bright-Line), and $3 \times 10^4$ cells were plated into 24-well plates 48 hr prior to infection.

To plate macrophages for infection, BMDM aliquots were thawed on ice, diluted into 9 ml of DMEM, centrifuged at 290 g for 5 min in an Eppendorf 5810 R centrifuge, and the pellet was resuspended in 10 ml BMDM media without antibiotics. $5 \times 10^5$ cells were plated into 24-well plates. Approximately 16 hr later, '30 % prep' *R. parkeri* were thawed on ice and diluted into fresh BMDM media to either $10^6$ p.f.u./ml or $2 \times 10^5$ p.f.u./ml. Media was then aspirated from the BMDMs and replaced with 500 µl media containing *R. parkeri*, and plates were spun at 300 g for 5 min in an Eppendorf 5810 R centrifuge. Infected cells were incubated in a humidified CEDCO 1600 incubator set to 33 °C and 5 % $CO_2$. Recombinant mouse IFN-β (PBL, 12405–1) was added directly to infected cells after infection.

For measuring p.f.u., supernatants from infected BMDMs were aspirated, and each well was washed twice with 500 µl sterile milli-Q-grade water. After adding 1 ml of sterile milli-Q water to each well, macrophages were lysed by repeated pipetting. Serial dilutions of lysates were added to confluent Vero cells in 12-well plates. Plates were spun at 300 g using an Eppendorf 5810 R centrifuge for 5 min at room temperature and incubated at 33 °C overnight. At ~16 h.p.i., media was aspirated and replaced with 2 ml/well of DMEM containing 0.7 % agarose and 5 % FBS (GemCell). At ~6 d.p.i., 1 ml of DMEM containing 0.7 % agarose, 1 % FBS (GemCell), 200 µg/ml amphotericin B (Invitrogen, 15290–018), and 2.5 % neutral red (Sigma) was added to each well. Plaques were then counted after 24 hr.

Microscopy, LDH, and IFN-I experiments were performed as described (*Burke et al., 2020*).

## Animal experiments

Animal research was conducted under a protocol approved by the University of California, Berkeley Institutional Animal Care and Use Committee (IACUC) in compliance with the Animal Welfare Act and other federal statutes relating to animals and experiments using animals (Welch lab animal use protocol AUP-2016-02-8426). The University of California, Berkeley IACUC is fully accredited by the Association for the Assessment and Accreditation of Laboratory Animal Care International and adheres to the principles of the Guide for the Care and use of Laboratory Animals (*National Research Council, 2011*). Mouse infections were performed in a biosafety level two facility. All animals were maintained at the University of California, Berkeley campus, and all infections were performed in accordance with the approved protocols. Mice were between 8 and 20 weeks old at the time of initial infection. Mice were selected for experiments based on their availability, regardless of sex. The sex of mice used for survival after i.d. infection and raw data for mouse experiments is provided in the Source Data for each figure. A statistical analysis was not performed to predetermine sample size prior to initial experiments. Initial sample sizes were based on availability of mice and the capacity to process or measure samples within a given time. After the first experiment, a Power Analysis was conducted to determine subsequent group sizes. All mice were of the C57BL/6 J background, except for outbred CD-1 mice. All mice were healthy at the time of infection and were housed in microisolator cages and provided chow, water, and bedding. No mice were administered antibiotics or maintained on water with antibiotics. Experimental groups were littermates of the same sex that were randomly assigned to experimental groups. For experiments with DKO mice deficient in *Ifnar1* and *Ifngr1*, mice were immediately euthanized if they exhibited severe degree of infection, as defined by a core body temperature dropping below 90 °F or lethargy that prevented normal movement.

## Mouse genotyping

Single mutant *Tlr4*$^{-/-}$ (**Hoshino et al., 1999**, *Ifnar1*$^{-/-}$ **Müller et al., 1994**, *Ifngr1*$^{-/-}$ **Huang et al., 1993**), DKO *Ifnar1*$^{-/-}$;*Ifngr1*$^{-/-}$, and WT C57BL/6J mice were previously described and originally obtained from Jackson Laboratories. CD-1 mice were obtained from Charles River. For genotyping, ear clips were boiled for 15 min in 60 µl of 25 mM NaOH, quenched with 10 µl Tris–HCl pH 5.5, and 2 µl of lysate was used for PCR using SapphireAMP (Takara, RR350) and gene-specific primers. Primers used were: *Ifnar1* forward (F): CAACATACTACAACGACCAAGTGTG; *Ifnar1* WT reverse (R): AACAAAC-CCCCAAACCCCAG; *Ifnar1*$^{-/-}$ R: ATCTGGACGAAGAGCATCAGG; *Ifngr1* (F): CTCGTGCTTTACGG-TATCGC; *Ifngr1* (R): TCGCTTTCCAGCTGATGTACT; WT *Tlr4* (F): CACCTGATACTTAATGCTGGCTGT AAAAAG; WT *Tlr4* (R): GGTTTAGGCCCCAGAGTTTTGTTCTTCTCA; *Tlr4*$^{-/-}$ (F): TGTTGCCCTTCA GTCACAGAGACTCTG; and *Tlr4*$^{-/-}$ (R): TGTTGGGTCGTTTGTTCGGATCCGTCG.

## Mouse infections

For mouse infections, *R. parkeri* was prepared by diluting 30%-prep bacteria into cold sterile PBS on ice. Bacterial suspensions were kept on ice during injections. For i.d. infections, mice were anaesthetized with 2.5 % isoflurane via inhalation. The right flank of each mouse was shaved with a hair trimmer (Braintree CLP-41590), wiped with 70 % ethanol, and 50 µl of bacterial suspension in PBS was injected intradermally using a 30.5-gauge needle. Mice were monitored for ~3 min until they were fully awake. No adverse effects were recorded from anesthesia. For i.v. infections, mice were exposed to a heat lamp while in their cages for approximately 5 min and then each mouse was moved to a mouse restrainer (Braintree, TB-150 STD). The tail was sterilized with 70 % ethanol, and 200 µl of bacterial suspension in sterile PBS was injected using 30.5-gauge needles into the lateral tail vein. Body temperatures were monitored using a rodent rectal thermometer (BrainTree Scientific, RET-3).

For fluorescent dextran experiments, mice were intravenously injected with 150 µl of 10 kDa dextran conjugated with Alexa Fluor 680 (D34680; Thermo Fisher Scientific) at a concentration of 1 mg/ml in sterile PBS (**Glasner et al., 2017**). As a negative control, mice with no *R. parkeri* infection were injected with fluorescent dextran. As an additional negative control, uninfected mice were injected intravenously with PBS instead of fluorescent dextran. At 2 hr post-injection, mice were euthanized with $CO_2$ and cervical dislocation, doused with 70 % ethanol, and skin surrounding the injection site (approximately 2 cm in each direction) was removed. Connective tissue between the skin and peritoneum was removed, and skin was placed hair-side-up on a 15 cm Petri dish. Skin was imaged with an LI-COR Odyssey CLx (LI-COR Biosciences), and fluorescence was quantified using ImageStudioLite v5.2.5. The skin from mice with no injected fluorescent dextran was used as the background measurement. Skin from mice injected with fluorescent dextran but no *R. parkeri* was normalized to an arbitrary number (100), and *R. parkeri*-infected samples were normalized to this value (*R. parkeri*-infected/uninfected × 100). The number of pixels at the injection site area was maintained across experiments (7,800 for small area and 80,000 for the large area).

All mice in this study were monitored daily for clinical signs of disease throughout the course of infection, such as hunched posture, lethargy, scruffed fur, paralysis, facial edema, and lesions on the skin of the flank and tail. If any such manifestations were observed, mice were monitored for changes in body weight and temperature. If a mouse displayed severe signs of infection, as defined by a reduction in body temperature below 90 °F or an inability to move normally, the animal was immediately and humanely euthanized using $CO_2$ followed by cervical dislocation, according to IACUC-approved procedures. Pictures of skin and tail lesions were obtained with permission from the Animal Care and Use Committee Chair and the Office of Laboratory and Animal Care. Pictures were captured with an Apple iPhone 8, software v13.3.1.

For harvesting spleens and livers, mice were euthanized at the indicated pre-determined times and doused with ethanol. Mouse organs were extracted and deposited into 50 ml conical tubes containing 4 ml sterile cold PBS for the spleen and 8 ml PBS for the liver. Organs were kept on ice and were homogenized for ~10 s using an immersion homogenizer (Fisher, Polytron PT 2500E) at ~22,000 r.p.m. Organ homogenates were spun at 290 g for 5 min to pellet the cell debris (Eppendorf 5810 R centrifuge). Twenty microliters of organ homogenates was then serial diluted into 12-well plates containing confluent Vero cells. The plates were then spun at 260 g for 5 min at room temperature (Eppendorf 5810 R centrifuge) and incubated at 33 °C. To reduce the possibility of contamination, organ homogenates were plated in duplicate and the second replicate was treated with 50 µg/ml carbenicillin

(Sigma) and 200 µg/ml amphotericin B (Gibco). The next day, at approximately 16 h.p.i., the cells were gently washed by replacing the existing media with 1 ml DMEM containing 2 % FBS (GemCell). The media were then aspirated and replaced with 2 ml/well of DMEM containing 0.7 % agarose, 5 % FBS, and 200 µg/ml amphotericin B. When plaques were visible at 6 d.p.i., 1 ml of DMEM containing 0.7 % agarose, 1 % FBS, and 2.5 % neutral red (Sigma) was added to each well, and plaques were counted at 24 h.p.i.

## Histology

For histology, spleens and livers were harvested from infected mice and immediately stored in 10 % neutral buffered formalin (Sigma HT501128). Histology was performed by HistoWiz Inc (http://histowiz.com) using a standard operating procedure and fully automated workflow. Samples were processed, embedded in paraffin, and sectioned at 4 µm thickness. BOND Polymer Refine Detection (Leica Biosystems) was used according to manufacturer's protocol. After staining with hematoxylin and eosin, sections were dehydrated and film coverslipped using a TissueTek-Prisma and Coverslipper (Sakura). Whole slide scanning (40 ×) was performed on an Aperio AT2 (Leica Biosystems). Histological consultation was blindly performed by a pathologist. Immunohistochemistry was performed with a rabbit anti-*Rickettsia* I7205 antibody (1:500 dilution; gift from Ted Hackstadt). Shareable links for all histology and immunohistochemistry data are available upon request to the authors.

## Statistical analysis

Statistical parameters and significance are reported in the figure legends. For comparing two sets of data, a two-tailed Student's T test was performed. For comparing two sets of in vivo p.f.u. data, Mann–Whitney U tests were used. For comparing two survival curves, log-rank (Mantel–Cox) tests were used. For comparing curves of two samples (mouse health, weight, and temperature), two-way ANOVAs were used. For two-way ANOVAs, if a mouse was euthanized prior to the statistical end point, the final value that was recorded for the mouse was repeated until the statistical endpoint. For two-way ANOVAs, if a measurement was not recorded for a timepoint, the difference between values at adjacent time points was used. Data were determined to be statistically significant when $p < 0.05$. Asterisks denote statistical significance as: $*p < 0.05$; $**p < 0.01$; $***p < 0.001$; $****p < 0.0001$, compared to indicated controls. Error bars indicate standard deviation (SD) for in vitro experiments and standard error of the mean (SEM) for in vivo experiments. All other graphical representations are described in the figure legends. Statistical analyses were performed using GraphPad PRISM v7.0.

## Acknowledgements

We thank Neil Fisher for editing this manuscript. PE was supported by postdoctoral fellowships from the Sweden-America Foundation. MDW was supported by NIH/NIAID grants R01AI109044 and R21AI138550. DRG, DAE, and EH were partially supported by NIH/NIAID grant R01 AI24493 (EH).

## Additional information

### Funding

| Funder | Grant reference number | Author |
| --- | --- | --- |
| National Institute of Allergy and Infectious Diseases | R01AI109044 | Matthew D Welch |
| National Institute of Allergy and Infectious Diseases | R21AI138550 | Matthew D Welch |
| National Institute of Allergy and Infectious Diseases | R01AI124493 | Eva Harris |

The funders had no role in study design, data collection and interpretation, or the decision to submit the work for publication.

## Author contributions

Thomas P Burke, Conceptualization, Data curation, Formal analysis, Investigation, Methodology, Supervision, Validation, Visualization, Writing – original draft, Writing – review and editing; Patrik Engström, Resources, Writing – review and editing; Cuong J Tran, Investigation, Writing – review and editing; Ingeborg M Langohr, Formal analysis, Investigation, Writing – review and editing; Dustin R Glasner, Diego A Espinosa, Investigation, Resources; Eva Harris, Resources, Supervision; Matthew D Welch, Conceptualization, Funding acquisition, Project administration, Supervision, Writing – review and editing

## Author ORCIDs

Thomas P Burke (ID) http://orcid.org/0000-0002-1385-8757
Patrik Engström (ID) http://orcid.org/0000-0003-3092-8216
Dustin R Glasner (ID) http://orcid.org/0000-0001-9821-6683
Diego A Espinosa (ID) http://orcid.org/0000-0002-4364-5031
Eva Harris (ID) http://orcid.org/0000-0002-7238-4037
Matthew D Welch (ID) http://orcid.org/0000-0003-2537-6349

## Ethics

Animal research was conducted under a protocol approved by the University of California, Berkeley Institutional Animal Care and Use Committee (IACUC) in compliance with the Animal Welfare Act and other federal statutes relating to animals and experiments using animals (Welch lab animal use protocol AUP-2016-02-8426). The University of California, Berkeley IACUC is fully accredited by the Association for the Assessment and Accreditation of Laboratory Animal Care International and adheres to the principles of the Guide for the Care and use of Laboratory Animals. Mouse infections were performed in a biosafety level 2 facility. All animals were maintained at the University of California, Berkeley campus, and all infections were performed in accordance with the approved protocols. Mice were immediately euthanized if they exhibited severe degree of infection, as defined by a core body temperature dropping below 90 F or lethargy that prevented normal movement.

## Decision letter and Author response

Decision letter https://doi.org/10.7554/eLife.67029.sa1
Author response https://doi.org/10.7554/eLife.67029.sa2

---

# Additional files

## Supplementary files

• Transparent reporting form

## Data availability

All data sets generated or analyzed during this study are included in the manuscript and supporting files.

---

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
