## [Decision Letter]

[Editors' note: this paper was reviewed by Review Commons.]

**Acceptance summary:**

This manuscript provides a murine model of *Rickettsia parkeri* infection, specifically for eschar formation upon rickettsiae infection. The researchers employed employed an intradermal (i.d.) infection model for infection using mice that were deficient in two interferon receptors (*Ifnar^-/-^Ifngr^-/-^*). Their study further describe new molecular mechanisms for rickettsiae infection during eschar formation, thereby, advancing our understanding on Rickettsia-Host Cell interaction.

---

## [Author Response]

We thank the *eLife* editors for inviting this resubmission and the *Review Commons* reviewers for their thoughtful comments and suggestions on how to improve the manuscript. We also thank the reviewers for describing the study as “highly significant,” “rigorous and reliable as described and can be reproduced by others,” and as “relevant to investigators working in the field of rickettsial diseases and to a broader audience studying mechanisms of intracellular parasitism and host responses.”

As proposed in our revision plan and suggested by all three reviewers, we have now performed histology in infected wild type and in interferon receptor-deficient mice at multiple time points of multiple organs. We also performed immunohistochemistry to identify the infected cell types in the skin, brain, lung, spleen and liver. Importantly, our findings are consistent with the histological findings observed in human eschars and with the observed infection of neutrophils and macrophages in *R. parkeri*-infected non-human primates and in mice. We feel these experiments have improved the manuscript and added to the strong significance of our findings to the field. We have also addressed all of the additional minor points raised by the reviewers.

Reviewer #1 (Evidence, reproducibility and clarity (Required)):“Rickettsial eschars are hallmarks of less severe spotted fever diseases. The underlying mechanisms involved in the formation of the eschar caused by pathogenic rickettsiae remains unknown. The authors of this manuscript studied this interesting research question by using Ifnar-/-Ifngr-/- mice and Sca2 or OmpB mutant of R. parkeri. R. parkeri probably is the best rickettsial species to study rickettsial eschar due to the clinical features of R. parkeri rickettsioses and the biosafety level required to work with it. The data presented in the manuscript are very promising. The conclusions are supported by the presented results. For the first time, this study recapitulated human eschar-like skin lesion observed in patients with R. parkeri rickettsioses in the mouse models. More interestingly, mice inoculated with Sca2 mutant of R. parkeri i.d. had less disseminated rickettsiae in tissues, which helps us to understand the mechanisms by which pathogenic rickettsiae cause systemic infection after the arthropod bite.”Minor comments:“(1) Figure 2D, it looks likely the lethality of mice i.d. infection with R. parkeri is not dose dependent. For example, mice inoculated with 10^4^ showed greater lethality compared to 10^7^. The authors might want to explain it in the Discussion.”

The reviewer is correct in observing that the lethality between different doses of *R. parkeri* in *Ifnar^-/^Ifngr^-/-^* mice after intradermal infection is not dose dependent with the current number of mice used per group. We do not understand the reason for this, and more broadly we do not understand the mechanism of lethality. We speculate that there could be a bottleneck that limits bacterial burdens; however, answering this question will require future investigations into the mechanisms of lethality that we feel are beyond the scope of this study. To address the reviewer’s point, we now include this statement: “Degrees of lethality between different doses in *Ifnar*^-/-^*Ifngr^-/-^* mice were not significantly different from one another, and the cause of lethality in this model remains unclear.”

“(2) Line 202, innate immunity in vitro might need to be revised.”

We agree that the previous description was vague. We changed the description to be more specific and it now reads: “…Sca2 does not significantly enhance the ability of *R. parkeri* to evade interferon-stimulated genes or inflammasomes in vitro*.*”

“(3) It is unclear what is the unit of the inoculum in animal experiments, PFU?”

Yes, it is PFU. We have now indicated this in the figure legends.

“(4) Line 36, in the study of "Reference 16", C3H/HeN mice, not B6 mice, were used.”

We thank the reviewer for noticing this error and we have changed the text to C3H/HeN.

“(5) The conclusion on eschar will be greatly strengthened if histological analysis is included, particularly whether dermis necrosis is present or not.”

In the revised manuscript we include the results of histology on the skin of wild type and *Ifnar^-/^Ifngr^-/-^* mice at 4, 7, and 12 d post infection (d.p.i.). We also used immunohistochemistry to identify the infected cell types. Images were blindly analyzed by Dr. Inge Langohr, a pathologist with expertise in characterizing rickettsial lesions. These experiments revealed significant epidermal and dermal necrosis, especially at 12 d.p.i., similar to what has been described in *R. parkeri-*infected humans and non-human primates.

“(6) Line 357, it is not clear what "spinfection" means.”

We have changed this to “infection” for clarity.

“Reviewer #1 (Significance (Required)):Several approaches employed in the study are new to the field of animal models of the rickettsioses. For example, fluorescent dextran was used to investigating the vascular damage in skin at the inoculation site; body temperature for mice infected with R. parkeri. Overall, the study is highly significant since it has answered the important questions in the research area of spotted fever rickettsioses and employed appropriate approaches. No major concerns were noticed.”

We thank the reviewer for appreciating the significance of this work.

Referees cross commentingI agree with other reviewers' comments. Thanks for the invite.”Reviewer #2 (Evidence, reproducibility and clarity (Required)):“The manuscript utilizes a new model of spotted fever rickettsiosis. Using this model, the authors have determined that knockout of the sca2 or ompB gene attenuates Rickettsia parkeri, and vaccination with the attenuated rickettsiae provides protection against virulent challenge. However, the model is far less than ideal as it has eliminated important effectors of immunity.”

We thank the reviewer for their comments, and we hope to have thoroughly addressed their concerns. In regard to the effects of interferons on long-lasting immunity to *R. parkeri,* we note to the reviewer that we observed that immunized *Ifnar^-/-^Ifngr^-/-^* mice were completely and robustly protected from rechallenge. No lethality and no loss of body weight or temperature was observed after a rechallenge dose of 10x the LD-50. These data reveal that interferons are dispensable for long-lasting immunity to *R. parkeri* in inbred mice and are not important effectors of adaptive immunity to *R. parkeri*. This is thus the first model that can be used to investigate the factors required for adaptive immunity to *R. parkeri* in mice.

If the reviewer’s comment is not referring to long-lasting adaptive immunity to *R. parkeri* but is instead referring to the general concept of using immunocompromised mice as animal models, we note that immunocompromised mice are used as models for a variety of pathogens, including many *Rickettsia* species (reviewed in Osterloh, *Med Microbiol Immunol* 2017), and *Ifnar^-/-^Ifngr^-/-^* mice specifically are used as models for Zika and Dengue virus infections. Unlike many other immunocompromised mice, *Ifnar^-/-^Ifngr^-/-^* mice do not require maintenance on antibiotics and they have no noticeable differences to wild type mice in regard to breeding or growth, thus they are well-suited as an inbred model for *R. parkeri*.

“Manuscript also fails to recognize that there is a Amblyomma maculatum tick transmitted model of Rickettsia parkeri infection that causes an eschar and disseminated pathology”.

We believe the reviewer is referring to Saito *et al.*, 2019, and we thank the reviewer for bringing this work to our attention. We had previously not acknowledged this work because we were unaware of it, as the paper did not initially appear in our PubMed searches. We now cite and discuss this work and compare our findings in light of the previous findings.

If the reviewer is instead referring to Banajee *et al.,* 2015, in the previous version of the manuscript in lines 266-269 we cited and acknowledged this reported tick transmission model in non-human primates.

As noted by Reviewer 3, our model with needle inoculation is significantly less time consuming and expensive than a tick transmission model. The eschars we observe in *Ifnar^-/-^Ifngr^-/-^* mice appear more similar to human eschars than those reported in non-human primates (Banajee et al) or any other mouse model*.* Moreover, needle inoculation makes it feasible to precisely measure the number of bacteria that are administered, which is not true with ticks. Lastly, our model reveals eschars in the C57BL/6 mouse strain, which have many genetic mutants available. Thus, our model provides many significant advantages over the tick model in C3H/HeN mice or non-human primates, including cost, time, reproducibility, availability of genetic mutants, and recapitulating lesions similar to human eschars.

“The model that they have used is inadequately characterized. The cutaneous lesion was not evaluated histologically to determine if it features the actual characteristics of an eschar.”

We thank the reviewer for this suggestion. As mentioned above in the response to reviewer #1, we have now performed histology and immunohistochemistry on the skin as well as on lungs and brain of WT and *Ifnar^-/-^Ifngr^-/-^* mice at 4, 7, and 12 dpi following i.d. infection with different doses of *R. parkeri*. We also performed a similar analysis of the spleens and livers after i.v. infection.

“Although bacteria were found in the liver and spleen, in which macrophages are significant target, there was no evaluation of the vital organs including lung and brain nor demonstration of the target cells or pathologic lesions.”

As mentioned above, we have now performed histology and immunohistochemistry of the skin, brain, and lung to identify the infected cell types after i.d. infection. We also performed these analyses with spleens and lungs after i.v. infection. Regardless of the administrative route or the organ, neutrophils and macrophages are the primary cell types with rickettsial immunostaining. In one tissue section in the lung, bacteria appeared to be in endothelial cells.

In previous work from our lab (Engström *et al.*, 2019), we found that lungs of wild type mice contained similar number of infectious *R. parkeri* as the spleen and liver after intravenous infection. Thus, in order to be able to process more samples quickly, we did not include lungs in the experiments in which plaque forming units were quantified. In unreported data, we also found that organs including the brain, kidneys, and heart had few recoverable PFUs.

“Unfortunately, the assay of vascular permeability was applied only to the inoculation site and not to the disseminated visceral organs such as lung and brain.”

We have performed the vascular permeability assay using internal organs alongside the skin; however, little/no fluorescence was observed in any sample. We were unable to distinguish differences between control groups or between control and experimental groups in organs from mice that were treated and untreated with the fluorescent dextran. Thus, we were unfortunately not able to apply the described vascular damage assay to organs other than the skin. We now indicate this in the revised text.

Reviewer #2 (Significance (Required)):“The authors all have misrepresented the eschar as a critically important lesion whereas the patients usually do not even know i's presence until they began to develop systemic symptoms and it is a detected by a physician examining the patient.”

We did not intend to suggest that the eschar is either more or less critically important than other features of rickettsial disease. We simply described the eschar as a “hallmark feature” of eschar-associated rickettsiosis. Additionally, as the reviewer notes, patients report systemic symptoms, and our model elicits systemic disease by *R. parkeri* in mice. Thus, the model we describe recapitulates both of the disease manifestations mentioned by the reviewer.

“On line 30 the authors state that mice are the natural reservoir of Rickettsia parkeri. The references cited describe the failure of acquisition by feeding ticks, meaning that it is not a true reservoir. The reference describing animals with antibodies merely indicates exposure to a spotted fever groupRickettsia not sufficient evidence of a role as a reservoir.”

We thank the reviewer for making this important distinction and we have altered the text to read:

“…small rodents including mice have been found as seropositive for *R. parkeri* in the wild.”

“In response to the request for my expertise, I have contributed a large amount of data to understanding mechanisms of immunity to rickettsiae and have developed several useful animal models of Rickettsial diseases. I also have expertise on clinical aspects of spotted fever group rickettsioses, including the eschar.”Referees cross commenting“This is not the first Mouse model of rickettsiosis to contain an eschar. There is a model of Rickettsia parkeri transmitted by Amblyomma maculatum ticks in which eschars occur.”

As noted above by us and also by Reviewer 3, we cited and discussed the tick transmission model in non-human primates (Banajee *et al.*, 2015) in the Discussion. We also now cite and discuss Saito *et al.,* 2019. We note that the lesions we observe in *Ifnar^-/-^Ifngr^-/-^* appear more necrotic and of similar size as the human eschars than those in reported in macaques in previous reports. Lastly, we note to the reviewer the many advantages of our i.d. infection model, including how it will make these experiments more widely accessible, more reproducible, less expensive, faster, will enable the infection of mice with various genetic modifications, and results in a skin lesion most similar to the human lesion.

Reviewer #3 (Evidence, reproducibility and clarity (Required)):“This manuscript reports novel observations pertinent to development in inbred mice of an eschar lesion and generalized lethal infection following intradermal infection with Rickettsia parkeri, the mice are deficient in two types of interferon receptors. This is a new observation for the murine system and expands the existing repertoire of model infections for tick-borne rickettsiae. This study also reports that Sca2-mediated actin-based motility is required for R. parkeri dissemination and provides indirect evidence that OmpB protein is involved in eschar formation, thus corroborating previous knowledge about these major surface exposed antigens of rickettsiae and host cell interactions and host responses to these organisms.”“The study is rigorous and reliable as described and can be reproduced by others given availability of adequate funding, access to similar facilities, strains of mice and rickettsial mutants, and technical personnel with similar skills and training. There are no ethical or technical concerns.”

We thank the reviewer for their thoughtful comments and for appreciating the advantages of this model.

“The main limitation of the manuscript is due to the fact that histological and immunohistochemical analysis of the eschar was not performed; therefore, it is not clear if pathological processes and features of this lesion formation are the same or related to the human pathology.”

We thank the reviewer for this suggestion. As mentioned above in the responses to Reviewers #1 and #2, we have now performed histological and immunohistochemical analyses of the skin and internal organs and had the images blindly analyzed by Dr. Inge Langohr, a pathologist who is an expert in lesions elicited by rickettsial pathogens. The features of the lesions, including necrosis, inflammation, and infected cell types, are similar to those observed in *R. parkeri-*infected humans and non-human primates.

“Similarly, in an attempt to generalize (as the authors try very hard), it is not clear how these observations will be relevant to rickettsial pathogens which are responsible for more severe forms of rickettsioses (such as R. rickettsii and R. prowazekii) but are not known to cause eschar formation as a part of their clinical manifestations.”

Our findings with Sca2 are in agreement with findings in *R. rickettsii* Sca2 in guinea pigs (Kleba *et al.*, 2010), which showed that Sca2 was required for eliciting fever and an antibody response. Our work also expands on these findings by showing that *sca2* mutants immunize against rechallenge and by finding reduced bacterial burdens in internal organs after intradermal infection with *sca2* mutant bacteria. Thus, we believe that studying *R. parkeri* genes in *Ifnar^-/-^Ifngr^-/-^* mice can serve as a model to better understand conserved virulence genes in diverse rickettsial pathogens.

Beyond virulence genes, we note that our model also recapitulates systemic disease including dissemination to internal organs. Thus, it provides a platform to study disease manifestations beyond the eschar that may be relevant to other rickettsial pathogens including *R. rickettsii* and *R. prowazekii*.

Some other virulent rickettsial pathogens cause limited/no disease in WT C57BL/6 mice, including *R. akari, R. conorii, R. typhi,* and *O. tsutsugamushi* (reviewed in Osterloh, *Med Microbiol Immunol* 2017). Thus, *Ifnar^-/-^Ifngr^-/-^* mice may potentially serve as models for these pathogens. We now include this point in the Discussion.

“The other deficiency is due to a limited description of the Sca2 and OmpB mutants used in this study. It was necessary to locate and review previous publications by this group in order to understand the experiments conducted here and their interpretation. It would be useful to the readers if this information (a better more complete description of the mutants and their properties) is summarized in this manuscript.”

We have now provided a more complete description of the mutants in the Introduction and Results.

Reviewer #3 (Significance (Required)):“The study is relevant to investigators working in the field of rickettsial diseases and to a broader audience studying mechanisms of intracellular parasitism and host responses.The study argues that difference(s) in dermal IFN signaling mechanism(s) distinguish human and murine susceptibility to R. parkeri infection. This is a very useful speculation; however, a better and deeper discussion would be helpful to demonstrate the relevance of these observations and their connection(s) to events occurring during the course of human infections. Regrettably, there are almost no citations of classic or current literature addressing these aspects of rickettsial pathogenesis and the role of IFN-dependent mechanisms beyond self-citations. Overall, the discussion includes four relatively short paragraphs, each addressing different directions of possible research, which indicates ample possible utility of this murine model; however, a more coherent and convincing discussion is desirable.”

We thank the reviewer for the suggestion, and we have now expanded the Discussion to address the role for IFN-dependent mechanisms in humans and mice during rickettsial infections, including classic and current literature citations.

Referees cross commenting“I agree with the Reviewer #2 that per se this is not the first murine model reproducing eschar upon A. maculatum transmission; however, this is the first model that allows to monitor eschar formation using needle inoculation. This model can be widely used; while many labs maybe limited by their facility setup and can't afford/conduct tick transmission experiments. The authors acknowledged existing of the tick transmission model and discuss inclusion of this option in their future experiments.”

We thank the reviewer for recognizing the many advantages of this model.